# Dual targeting of histone deacetylases and MYC as potential treatment strategy for H3-K27M pediatric gliomas

Danielle Algranati[1†], Roni Oren[2†], Bareket Dassa[3], Liat Fellus-Alyagor[2], Alexander Plotnikov[4], Haim Barr[4], Alon Harmelin[2], Nir London[5], Guy Ron[6], Noa Furth[1]*, Efrat Shema[1]*

[1]Department of Immunology and Regenerative Biology, Weizmann Institute of Science, Rehovot, Israel; [2]Department of Veterinary Resources, Weizmann Institute of Science, Rehovot, Israel; [3]Bioinformatics Unit, Department of Life Sciences Core Facilities, Faculty of Biochemistry, Weizmann Institute of Science, Rehovot, Israel; [4]Wohl Institute for Drug Discovery of the Nancy and Stephen Grand Israel National Center for Personalized Medicine, Weizmann Institute of Science, Rehovot, Israel; [5]Department of Chemical and Structural Biology, Weizmann Institute of Science, Rehovot, Israel; [6]Racah Institute of Physics, Hebrew University, Jerusalem, Israel

*For correspondence:
noa.furth@weizmann.ac.il (NF);
efrat.shema@weizmann.ac.il (ES)

[†]These authors contributed equally to this work

**Abstract** Diffuse midline gliomas (DMGs) are aggressive and fatal pediatric tumors of the central nervous system that are highly resistant to treatments. Lysine to methionine substitution of residue 27 on histone H3 (H3-K27M) is a driver mutation in DMGs, reshaping the epigenetic landscape of these cells to promote tumorigenesis. H3-K27M gliomas are characterized by deregulation of histone acetylation and methylation pathways, as well as the oncogenic MYC pathway. In search of effective treatment, we examined the therapeutic potential of dual targeting of histone deacetylases (HDACs) and MYC in these tumors. Treatment of H3-K27M patient-derived cells with Sulfopin, an inhibitor shown to block MYC-driven tumors in vivo, in combination with the HDAC inhibitor Vorinostat, resulted in substantial decrease in cell viability. Moreover, transcriptome and epigenome profiling revealed synergistic effect of this drug combination in downregulation of prominent oncogenic pathways such as mTOR. Finally, in vivo studies of patient-derived orthotopic xenograft models showed significant tumor growth reduction in mice treated with the drug combination. These results highlight the combined treatment with PIN1 and HDAC inhibitors as a promising therapeutic approach for these aggressive tumors.

## eLife assessment

This work contributes to the study of H3-K27M mutated pediatric gliomas. It **convincingly** demonstrates that the concomitant targeting of histone deacetylases (HDACs) and the transcription factor MYC results in a notable reduction in cell viability and tumor growth. This reduction is linked to the suppression of critical oncogenic pathways, particularly mTOR signaling, emphasizing the role of these pathways in the disease's pathogenesis. The current version of the manuscript is **important** because it unveils a vulnerability from dual targeting HDACs and MYC in the context of pediatric gliomas. This work will be of interest to cancer epigenetics and therapeutics research, with a focus on the neuro-oncology field.

**eLife digest** Diffuse midline gliomas (DMGs) are among the most aggressive and fatal brain cancers in children. They are often associated with changes in histones, the proteins that control gene activity and give chromosomes their structure. Most children with DMGs, for example, share the same anomaly in their histone H3 protein (referred to as the H3-K27M mutation). This change affects how small chemical tags called methyl and acetyl groups can be added onto histone 3, which in turn alters the way the protein can switch genes on and off. As a result, tumours start to develop.

One potential therapeutic strategy against DMGs is to use histone deacetylase inhibitors (HDACi), a promising type of drugs which inhibits the enzymes that remove acetyl groups from histones. Patients can develop resistance to HDACi, however, highlighting the need to explore other approaches. One possibility is to treat patients with several types of drugs, each usually targeting a distinct biological process that contributes to the emergence of cancer. This combined approach can have multiple benefits; the drugs potentially amplify each other's effect, for example, and it is also less likely for cells to become resistant to more than one compound at the time. In addition, each drug in the combination can be used in a lower dose to reduce side effects and benefit patients.

DMG tumour cells often feature higher activity levels of a protein known as MYC, which can contribute to the growth of the tumour. Algranati, Oren et al. therefore set out to test whether combining an HDACi known as Vorinostat with a drug that blocks MYC activity (Sulfopin) can act as an effective treatment for this cancer.

Tumour samples from eight DMG patients were treated with either Sulfopin alone, or Sulfopin in association with Vorinostat. Cells exposed to both drugs were less likely to survive, and additional genetic experiments showed that the combined treatment had resulted in pathways that promote tumour development being blocked. When both Sulfopin and Vorinostat were administered to mice made to grow human DMG tumors, the animals showed a greater reduction in tumor growth.

Treatment options for DMG are usually limited, with chemotherapy often being ineffective and surgery impossible. The work by Algranati, Oren et al. suggests that combining HDACi and drugs targeting the MYC pathway is a strategy that should be examined further to determine whether clinical applications are possible.

## Introduction

Diffuse midline gliomas (DMGs) are fatal tumors of the central nervous system that occur primarily in children (*Johung and Monje, 2017*; *Hayden et al., 2021*). Years of clinical trials have revealed that conventional chemotherapy is ineffective, and with no option for complete resection, DMG is now the leading cause of brain tumor-related death in children (*Ostrom et al., 2015*; *Srikanthan et al., 2021*). Sequencing studies showed that up to 80% of the diagnosed children exhibit a lysine 27 to methionine (H3-K27M) mutation in one of the genes encoding histone H3.3 and H3.1 (clinically classified as DMG, H3 K27M-mutant). Numerous subsequent studies revealed pronounced alterations in the epigenetic landscape driven by this histone mutation (*Castel et al., 2015*; *Fontebasso et al., 2014*; *Schwartzentruber et al., 2012*; *Wu et al., 2012*). Among these global epigenetic alterations are a drastic loss of H3 lysine 27 tri-methylation (H3K27me3), concomitant with an increase in H3 lysine 27 acetylation (H3K27ac). These epigenetic alterations were shown to affect transcriptional programs and support tumorigenesis (*Bender et al., 2013*; *Chan et al., 2013*; *Lewis et al., 2013*; *Krug et al., 2019*; *Piunti et al., 2017*; *Harpaz et al., 2022*).

Detailed characterization of the modes of chromatin deregulation in these tumors pointed toward targeting epigenetic pathways as a promising therapeutic approach (*Filbin and Monje, 2019*). Specifically, pervasive H3K27 acetylation patterns observed in H3-K27M DMGs, along with enrichment of H3K27ac specifically on mutant nucleosomes (*Lewis et al., 2013*; *Krug et al., 2019*; *Piunti et al., 2017*; *Furth et al., 2022*), underscored the histone acetylation pathway as a potential drug target for this cancer. Indeed, pan-histone deacetylase inhibitors (HDACi), such as Vorinostat and Panobinostat, were found to inhibit growth and to restore gene expression alterations observed in H3-K27M malignant gliomas (*Grasso et al., 2015*; *Yin et al., 2007*; *Su et al., 2022*). However, recent studies demonstrated that H3-K27M cells develop resistance to HDACi treatment (*Grasso et al., 2015*; *Su*

*et al., 2022*), stressing the importance of identifying novel drug combinations to induce synergistic inhibition of oncogenic pathways.

The MYC oncogenic pathway is frequently deregulated in diverse cancers, and thus stands as a prominent therapeutic target (*Chen et al., 2018*). High-level gene amplification of *MYC* is recurrently observed in pediatric high-grade gliomas (*Buczkowicz et al., 2014*; *Mackay et al., 2017*). H3-K27M gliomas show high expression of *MYC* and MYC target genes, due to both epigenetic alterations and structural variants, resulting in a viability dependency on MYC signaling (*Krug et al., 2019*; *Pajovic et al., 2020*; *Dubois et al., 2022*). Unfortunately, directly targeting MYC in tumors is extremely challenging (*Wang et al., 2021*). *Dubiella et al., 2021* have recently developed a unique inhibitor targeting peptidyl-prolyl isomerase NIMA-interacting-1 (PIN1), an upstream regulator of MYC activation (). PIN1 itself is frequently upregulated in many types of cancers, resulting in sustained proliferative signaling and tumor growth (*Farrell et al., 2013*; *D'Artista et al., 2016*). The PIN1 inhibitor, Sulfopin, was shown to downregulate MYC target genes in vitro and to block MYC-driven tumors in murine models of neuroblastoma (*Dubiella et al., 2021*).

To address the increasing need for an effective combination therapy for H3-K27M DMGs, we tested whether dual targeting of histone acetylation and MYC activation, using Vorinostat and Sulfopin, would yield beneficial outcome. We show that the combined treatment substantially reduces viability of patient-derived glioma cells, in concordance with the inhibition of prominent oncogenic pathways, including the mTOR pathway. Treatment of H3-K27M DMG xenograft mice models with this drug combination led to reduced tumor growth and confirmed downregulation of mTOR in vivo. Together, these results suggest the combination of PIN1 and HDAC inhibitors as a promising therapeutic strategy for H3-K27M gliomas.

## Results

### Sulfopin inhibits MYC targets and reduces viability of H3-K27M glioma cells

In light of the important role of MYC in H3-K27M DMGs, we set to explore whether the PIN1 inhibitor, Sulfopin, may affect MYC signaling in these cells and thus confer therapeutic potential. Sulfopin-dependent inhibition of PIN1 was shown to downregulate the expression of MYC target genes in several cancer cell lines (*Dubiella et al., 2021*). Indeed, RNA sequencing (RNA-seq) analysis of H3-K27M-mutant patient-derived glioma cells (SU-DIPG13) treated with Sulfopin revealed strong enrichment of MYC targets among the downregulated genes (*Figure 1A*). The genes downregulated by Sulfopin were also enriched for MYC-bound genes identified in several non-glioma cell lines, supporting a role for PIN1 in activation of the MYC pathway also in these glioma cells (*Figure 1—figure supplement 1A and B*). Of note, Sulfopin was developed as a covalent inhibitor of PIN1, and did not lead to PIN1 degradation, as previously shown (*Figure 1—figure supplement 1C*; *Dubiella et al., 2021*). We further validated the decreased expression of prominent MYC targets (*Farrell et al., 2013*; *Hölzel et al., 2005*; *Teng et al., 2004*) upon Sulfopin treatment by RT-qPCR analysis (*Figure 1B* and *Figure 1—figure supplement 1D*). Interestingly, Cut&Run analysis (*Skene and Henikoff, 2017*) revealed an increase in H3K27me3 levels, indicative of repressed chromatin (*Ferrari et al., 2014*), around the transcriptional start sites (TSS) of MYC target genes, correlating with silencing of these genes upon PIN1 inhibition (*Figure 1C* and *Figure 1—figure supplement 1E*). Of note, global H3K27me3 distribution on all TSS and gene body regions was not affected by Sulfopin treatment, supporting a specific increase in H3K27me3 on the TSS of MYC target genes (*Figure 1—figure supplement 1F and G*). Furthermore, Sulfopin-unique H3K27me3 peaks (i.e. peaks that were found only in the Sulfopin-treated cells, and not in DMSO-treated cells) were enriched for MYC target genes (*Figure 1D*).

We next explored the effect of Sulfopin on the viability of eight patient-derived DMG cultures, harboring either the H3.3-K27M or H3.1-K27M mutation. Inhibition of PIN1 by Sulfopin was previously shown to delay cell cycle progression and thus reduce cell viability after extended treatments of 6–8 days (*Dubiella et al., 2021*). In line with downregulation of MYC signaling, Sulfopin treatment led to a mild reduction in cell viability, conserved across the different cultures (*Figure 1E*). To examine whether H3-K27M mutation renders DMG cells more sensitive to Sulfopin treatment, we took advantage of two sets of isogenic H3-K27M-mutant patient-derived DMG cell lines (SU-DIPG13, BT245) in

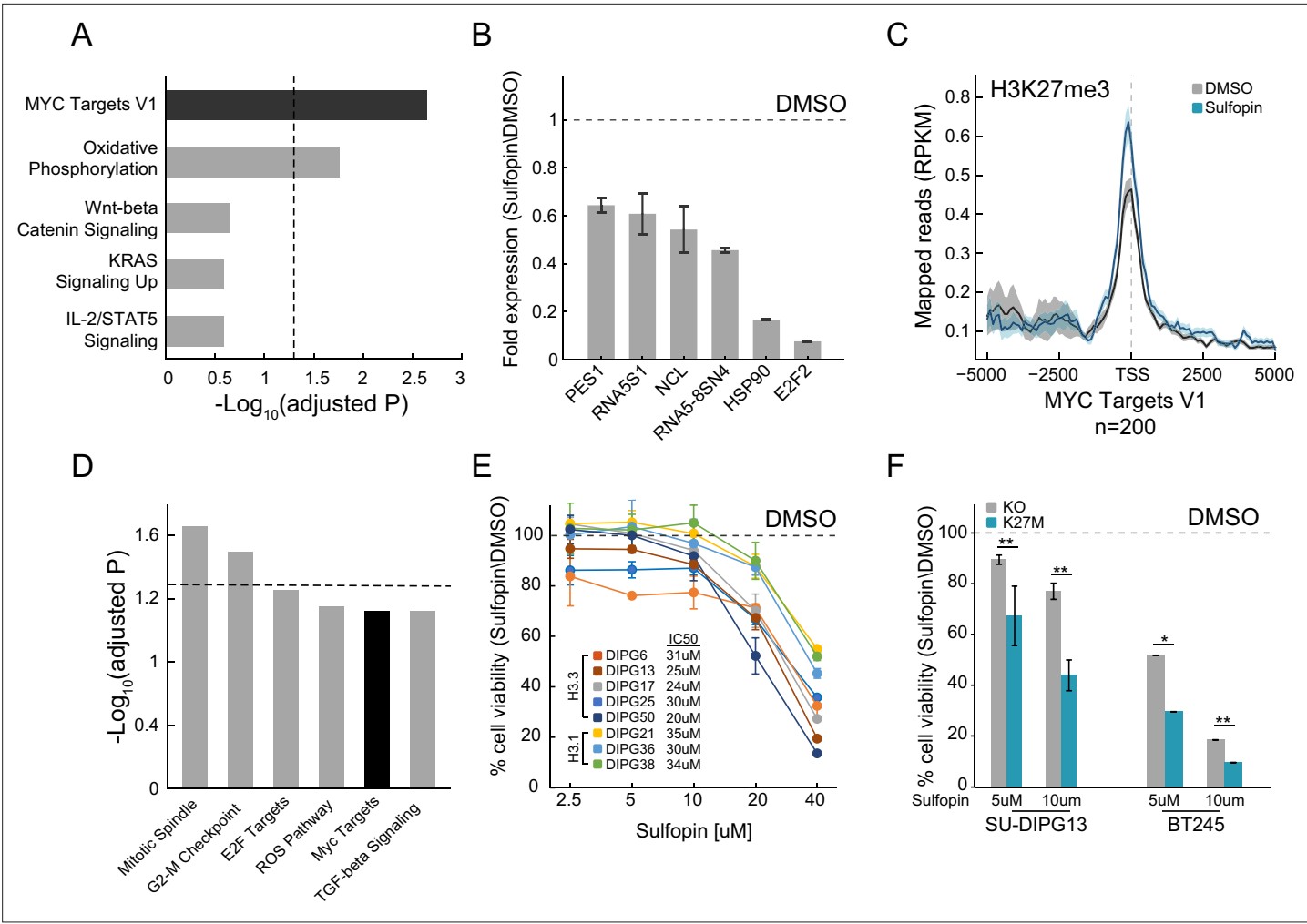

**Figure 1.** Sulfopin inhibits MYC signaling and reduces viability of diffuse midline glioma (DMG) cells in an H3-K27M-dependent manner. (**A**) Functional enrichment analysis on significantly downregulated genes in SU-DIPG13 cells treated with 10 µM Sulfopin for 12 hr, compared to DMSO. Enrichr algorithm (**Kuleshov et al., 2016**) was used to compare downregulated genes against the Molecular Signatures Database (MSigDB) hallmark gene set (**Liberzon et al., 2015**). Dashed line denotes adjusted p-value=0.05. MYC targets are significantly enriched among the Sulfopin downregulated genes. (**B**) RT-qPCR analysis of selected MYC target genes in SU-DIPG13 cells treated with 10 µM Sulfopin for 12 hr, compared to DMSO. Fold change between Sulfopin and DMSO-treated cells was calculated and the mean ± SD of two technical repeats is shown. (**C**) H3K27me3 Cut&Run read coverage over MYC target genes ('MYC Targets V1' hallmark gene set; **Liberzon et al., 2015**, n=200), in SU-DIPG13 cells treated with 10 µM Sulfopin for 8 days compared to DMSO. Sulfopin treatment increases H3K27me3 levels on the transcriptional start sites (TSS) of MYC target genes. (**D**) Functional enrichment analysis of the genes associated with Sulfopin-unique H3K27me3 peaks, in SU-DIPG13 cells treated as in C. Dashed line denotes adjusted p-value=0.05. MYC target gene set ('MYC Targets V1' hallmark gene set; **Liberzon et al., 2015**) is mildly enriched among these genes, with adjusted p-value of 0.077. (**E**) Cell viability, as measured by CellTiterGlo, of eight DMG cultures (H3.3K27M: SU-DIPG13, SU-DIPG6, SU-DIPG17, SU-DIPG25, and SU-DIPG50. H3.1K27M: SU-DIPG36, SU-DIPG38, and SU-DIPG21), treated with Sulfopin for 8 days with pulse at day 4, compared to DMSO. Mean ± SD of two technical replicates is shown. Logarithmic scale is used for the x-axis. Sulfopin treatment led to a mild reduction in cell viability in all H3-K27M glioma cultures. (**F**) Cell viability, as measured by CellTiterGlo, of two isogenic DMG cell lines (SU-DIPG13 and BT245) in which the mutant histone was knocked out (KO), treated with the indicated concentration of Sulfopin for 8 days, compared to DMSO. For each cell line and concentration, the fold change in viability between Sulfopin and DMSO-treated cells is shown. For SU-DIPG13 mean±SE of at least two independent experiments is shown. For BT245 mean ± SD of three technical replicates is shown. H3-K27M glioma cells show higher sensitivity to Sulfopin treatment compared to the KO cells. *p<0.05; **p<0.01 (two-sample t-test over all technical replicates). Significance adjusted after Bonferroni correction.

The online version of this article includes the following source data and figure supplement(s) for figure 1:

**Figure supplement 1.** The sensitivity of diffuse midline glioma (DMG) cells to Sulfopin correlates with MYC expression levels.

**Figure supplement 1—source data 1.** Original scan for the western blot analysis shown in **Figure 1—figure supplement 1C**.

**Figure supplement 1—source data 2.** Annotation of the original scan for the western blot analysis shown in **Figure 1—figure supplement 1C**.

which the *H3F3A*-K27M-mutant allele has been knocked out (KO). In both lines, the reduction in cell viability was partially rescued in the KO cells (*Figure 1F*), suggesting higher sensitivity of H3-K27M-mutant cells to MYC inhibition by Sulfopin. These results are in line with previous reports showing H3-K27M-dependent activation of the MYC/RAS axis in mouse and human cells (*Krug et al., 2019*; *Pajovic et al., 2020*; *Dubois et al., 2022*). Of note, DMG cells showed greater sensitivity to the inhibitor compared to human astrocytic-like cells (*Figure 1—figure supplement 1H*).

To further explore the status of MYC and PIN1 as indicators of response to Sulfopin inhibition, we analyzed published transcriptomic dataset of DMG tumors (*Berlow et al., 2018*). Interestingly, while DMG samples showed significantly higher expression of MYC and its target genes compared to normal samples, PIN1 expression was lower in the cancer compared to matched normal tissue (*Figure 1—figure supplement 1I–K*). Furthermore, H3-K27M-mutant tumors showed higher expression of MYC, compared to H3 WT tumors, while PIN1 levels were reduced across samples (*Mackay et al., 2017*; *Figure 1—figure supplement 1L and M*). These results suggest that MYC expression levels, and not PIN1, may determine the sensitivity of H3-K27M cells to Sulfopin treatment. To examine this notion, we measured the expression levels of MYC, as well as the doubling time of the different DMG cultures. The response to Sulfopin treatment (as reflected by IC50 measurements) correlated with the cells' doubling time (*Figure 1—figure supplement 1N*). Cells that divide faster were more sensitive to Sulfopin treatment, in line with PIN1 role in regulating cell cycle progression, and specifically G2/M transition (*Yeh and Means, 2007*; *Pinch et al., 2020*). As expected, the doubling time negatively correlated with MYC expression, suggesting that MYC levels may predict the response to Sulfopin (*Figure 1—figure supplement 1O*). Indeed, MYC levels negatively correlated with the IC50 of Sulfopin; thus, cells relaying on high MYC activity showed higher sensitivity to Sulfopin (*Figure 1—figure supplement 1P*).

H3K27me3 levels were shown to be drastically deregulated in DMG cells expressing H3-K27M, with global loss of this modification across the genome, and retention of residual H3K27me3 at CpG islands (CGIs). These residual H3K27me3 levels were shown to be necessary to silence tumor suppressor genes and maintain cell proliferation in DMG cells (*Mohammad et al., 2017*; *Jain et al., 2020*). Interestingly, Sulfopin treatment resulted in reduction of H3K27me3 signal over CGIs, perhaps contributing to Sulfopin-dependent growth inhibition in these cells (*Figure 1—figure supplement 1Q*). Taken together, our results indicate that PIN1 inhibition by Sulfopin downregulated MYC target genes, reduced H3K27me3 levels at CGIs, and led to reduce viability of DMG cells in an H3-K27M-dependent manner.

## Combination of Sulfopin and Vorinostat elicits robust inhibition of oncogenic pathways and an additive effect on cell viability

Inspired by the positive outcome of combinatorial drug therapy in H3-K27M DMG models (*Grasso et al., 2015*; *Lin et al., 2019*), we aimed to explore whether combining Sulfopin treatment with an epigenetic drug will further reduce cell viability. H3-K27M gliomas are characterized by deregulation of epigenetic pathways affecting post-translational modifications of lysine 27 (methylation and acetylation) and lysine 4 (methylation) on histone H3 (*Bender et al., 2013*; *Chan et al., 2013*; *Lewis et al., 2013*; *Krug et al., 2019*; *Furth et al., 2022*; *Harutyunyan et al., 2019*). Thus, we explored the effects of combining sub-IC50 concentration of Sulfopin with four epigenetic inhibitors targeting critical components associated with these modifications: (1) EPZ-6438, an inhibitor of EZH2, the catalytic subunit of the PRC2 complex that deposits H3K27me3 (*Schuettengruber et al., 2017*). H3-K27M cells were reported to be sensitive to EZH2 inhibition as they rely on the residual H3K27me3 levels to suppress key tumor suppressor genes (*Piunti et al., 2017*; *Mohammad et al., 2017*; *Stafford et al., 2018*; *Jain et al., 2019*); (2) GSK-J4, an inhibitor of the H3K27me3 demethylase JMJD3 (*Agger et al., 2007*), shown to rescue H3K27me3 levels in H3-K27M gliomas (*Grasso et al., 2015*; *Hashizume et al., 2014*); (3) MM-102, an inhibitor of the H3K4me3 methyltransferase MLL1 (*Schuettengruber et al., 2017*), shown to reduce viability of H3-K27M-mutant cells (*Furth et al., 2022*); and (4) Vorinostat, an HDACi that was shown to have clinical benefit in H3-K27M gliomas (*Grasso et al., 2015*; *Yin et al., 2007*; *Su et al., 2022*). Treatment dosages of these drugs were set according to previous studies (*Mohammad et al., 2017*; *Grasso et al., 2015*; *Furth et al., 2022*). We found that the combination of Sulfopin and Vorinostat (HDACi) had the strongest effect, reducing viability by 80% (*Figure 2—figure supplement 1A*).

Next, we adjusted treatment protocols to account for the optimal treatment period for each drug, and examined the effect of this drug combination on the viability of eight H3-K27M-mutant patient-derived DMG cultures, across different dosages (*Figure 2A and B* and *Figure 2—figure supplement 1B–D*). For each pair of concentrations, we calculated a BLISS index; an additive effect of the two drugs was determined as BLISS index ~1 (*Yadav et al., 2015*; *Bliss, 1939*; *Figure 2C* and *Figure 2—figure supplement 1E*). While the combined treatment was effective in all the cultures, cells harboring H3.3-K27M showed higher sensitivity (*Figure 2D* and *Figure 2—figure supplement 1F*). Moreover, while an additive effect of Sulfopin and Vorinostat was detected in all the H3.3-K27M DMG cultures in most of the drug doses, for the H3.1-K27M cultures we measured an additive effect only in higher dosages of Vorinostat (*Figure 2C–E* and *Figure 2—figure supplement 1E and F*). In line with the difference in sensitivity, H3.1- and H3.3-K27M cells differed substantially in MYC expression levels, with H3.1-K27M cells expressing significantly lower levels of MYC (*Figure 2—figure supplement 1G*). Indeed, BLISS indexes negatively correlated with MYC expression levels (*Figure 2F*), indicating that H3.3-K27M cells, which express higher levels of MYC, are more sensitive to the combination of Sulfopin and Vorinostat. Importantly, this highlights MYC, and MYC target genes, as potential biomarkers of response for the combined treatment. For example, the expression levels of the two MYC targets, NOC4L and PSMD3, negatively correlated with the BLISS indexes and significantly differed between H3.1- and H3.3-K27M cells (*Figure 2F* and *Figure 2—figure supplement 1H*). Furthermore, these and additional MYC targets were downregulated following the combined treatment in the H3.3-K27M culture, SU-DIPG13 (*Figure 2—figure supplement 1I*). Overall, our data demonstrates that combining Sulfopin treatment with Vorinostat results in an additive effect on the viability of DMG cultures, with H3.3-K27M cells showing higher sensitivity. This is in line with previous studies highlighting clinical and molecular distinction between H3.1- and H3.3-K27M-mutant tumors, which is thought to dictate their targetable vulnerabilities (*Castel et al., 2015*; *Nagaraja et al., 2019*; *Jessa et al., 2022*).

To gain more insights on the molecular mechanisms underlying this effect on viability, we profiled the transcriptome of SU-DIPG13 cells (H3.3-K27M) treated with each of the drugs alone and the combined treatment (*Figure 2—figure supplement 1J*). A total of 494 genes were significantly differentially expressed (DE) in cells treated with the drug combination, compared to only 145 and 167 in cells treated solely with Sulfopin or Vorinostat, respectively (*Figure 2—figure supplement 1K and L*). Hierarchical clustering of all 620 DE genes confirmed strong additive effect of the combined treatment on gene expression, primarily demonstrated by clusters 1 and 4 (*Figure 2G*; *Supplementary file 1*). Cluster 1, consisting of approximately half of the DE genes, was strongly downregulated by the combined treatment, compared to single-agent treated cells. This cluster included genes that play crucial roles in glioma progression such as *MTOR*, *ATF3*, *CREB3*, and *TGFA* (*Annovazzi et al., 2009*; *Ma et al., 2015*; *Xue et al., 2016*; *Tang et al., 1997*; *Figure 2—figure supplement 1M*). Moreover, the mTORC1 signaling pathway was highly enriched within the genes comprising cluster 1 (*Figure 2—figure supplement 1N*). Importantly, gene set enrichment analysis (GSEA) revealed that this pathway was significantly enriched in downregulated genes following the combined treatment (*Figure 2H* and *Figure 2—figure supplement 1O*). Of note, additional epigenetic and oncogenic pathways such as BMI1, cAMP, and IL-6/JAK/STAT3 were significantly downregulated in cells treated with the drug combination (*Abdouh et al., 2009*; *Zhang et al., 2020*; *West et al., 2018*; *Figure 2I* and *Figure 2—figure supplement 1O*). Reduction in mTOR signaling (mTOR, phospho-mTOR, and phospho-S6; *Kennedy and Lamming, 2016*) and ATF3 was further validated by western blot and RT-qPCR analysis (*Figure 2J* and *Figure 2—figure supplement 1P*). Importantly, we also confirmed that the combined treatment with Sulfopin and Vorinostat led to induction of p21 (CDKN1A), a prominent tumor suppressor promoting cell cycle arrest and regulated by mTOR signaling (*Shamloo and Usluer, 2019*; *Tahmasebi et al., 2014*; *Cui et al., 2021*; *Figure 2H and J*). Finally, exploring mTOR expression in all the tested DMG cultures revealed a strong negative correlation with the BLISS indexes of the combined treatment (*Figure 2—figure supplement 1Q*), suggesting mTOR as a potential biomarker of response for the combined treatment of Sulfopin and Vorinostat. Overall, our data revealed that the combined inhibition of HDACs and MYC results in robust downregulation of prominent oncogenic pathways.

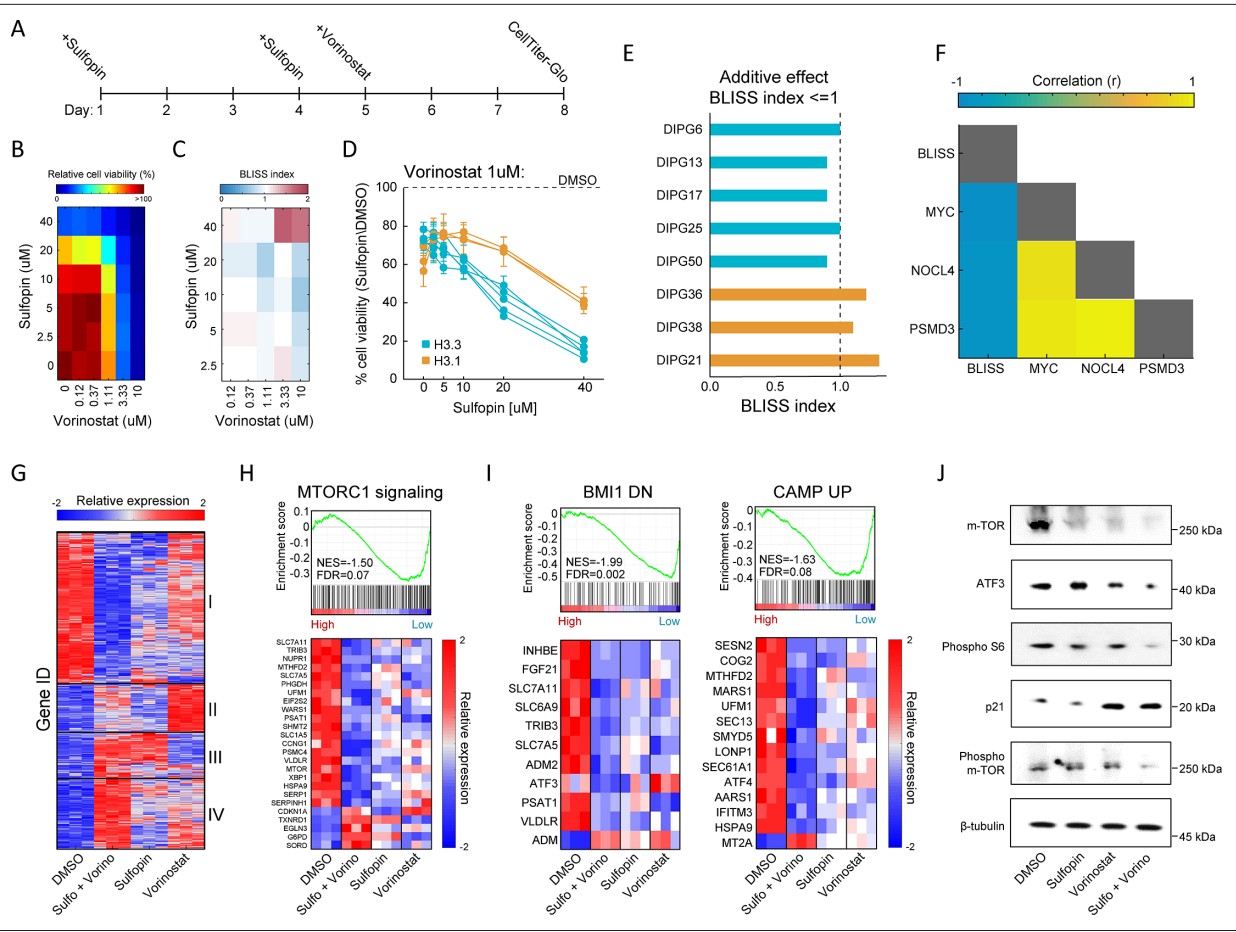

**Figure 2.** Combination of Sulfopin and Vorinostat elicits robust downregulation of oncogenic pathways. (**A**) Timeline demonstrating the treatment protocol for the combination of Sulfopin and Vorinostat. (**B**) Percentage of cell viability, as measured by CellTiterGlo of SU-DIPG13 cells treated with Sulfopin and Vorinostat at the indicated concentrations, compared to DMSO. (**C**) BLISS index measured as the ratio between the observed and the expected effect of the combination of Sulfopin and Vorinostat, for each pair of concentrations, in SU-DIPG13. Synergy: Bliss < 1, Additive: Bliss = 1, Antagonist: Bliss > 1. (**D**) Cell viability as measured by CellTiterGlo, of eight diffuse midline glioma (DMG) cultures treated with Sulfopin (0 µM, 2.5 µM, 5 µM, 10 µM, 20 µM, and 40 µM) and Vorinostat (1 µM), compared to DMSO. H3.3-K27M and H3.1-K27M cultures are indicated in blue and orange, respectively. Mean ± SD of two technical replicates is shown. H3.3-K27M cells showed higher sensitivity to the combined treatment compared to H3.1-K27M cells. (**E**) The BLISS index of the combination of Sulfopin (10 µM) and Vorinostat (1 µM), in the indicated cultures. An additive effect was detected in all the H3.3-K27M cultures at this set of concentrations. (**F**) Pearson correlation coefficient matrix of BLISS index of the combined treatment (Sulfopin [10 µM] and Vorinostat [1 µM]) and mRNA levels of MYC and its target genes, in the eight DMG cultures tested. mRNA levels were measured by RT-qPCR (*Figure 2—figure supplement 1G and H*). Blue and yellow colors indicate negative or positive correlation, respectively. Negative correlation was detected between the BLISS indexes and the expression levels of MYC and its target genes. (**G**) Unsupervised hierarchical clustering of expression levels of 620 significantly differentially expressed (DE) genes detected in SU-DIPG13 cells treated with either Sulfopin (10 µM, 8 days), Vorinostat (1 µM, 72 hr), the combination of Sulfopin and Vorinostat or DMSO (see also *Supplementary file 1*). Gene expression rld values (log2 transformed and normalized) were standardized for each gene (row) across all samples. Color intensity corresponds to the standardized expression, low (blue) to high (red). Clusters 1 and 4 demonstrate additive transcriptional patterns associated with the combined treatment. (**H**) Top: Gene set enrichment analysis (GSEA) on SU-DIPG13 treated with combination of 10 µM Sulfopin and 1 µM Vorinostat compared to DMSO, showing significant downregulation of mTORC1 signaling ('HALLMARK_MTORC1_SIGNALING' gene set; *Liberzon et al., 2015*) in the combined treatment. NES: normalized enrichment score. FDR: false discovery rate. Bottom: Expression levels of significantly DE genes detected in the combined treatment compared to DMSO that are part of the mTORC1 signaling gene set. *MTOR* gene was added manually to the heatmap. Heatmaps were generated as described in B. (**I**) Top: GSEA on SU-DIPG13 treated as in C, showing significant downregulation of the epigenetic BMI-1 pathway and the oncogenic cAMP pathway in the combined treatment (BMI1_DN.V1_UP; CAMP_UP.V1_UP; Molecular Signatures Database [MSigDB] C6 oncogenic signature; *Wiederschain et al., 2007*; *van Staveren et al., 2006*). Bottom: Expression levels of significantly DE genes detected in the combined treatment compared to DMSO that are part of the BMI-1 and cAMP gene sets. Heatmaps were generated as described B. (**J**) Western blot of SU-DIPG13 treated either with Sulfopin (10 µM, 8 days), Vorinostat (1 µM, 72 hr), the combination of Sulfopin and Vorinostat or DMSO, using the indicated antibodies. β-Tubulin is used as loading control.

The online version of this article includes the following source data and figure supplement(s) for figure 2:

*Figure 2 continued on next page*

*Figure 2 continued*

**Source data 1.** Original scan for the western blot analysis shown in *Figure 2J*.

**Source data 2.** Annotation of the original scan for the western blot analysis shown in *Figure 2J*.

**Figure supplement 1.** The additive effect of the combination of Sulfopin and Vorinostat results in downregulation of oncogenic pathways.

## Sulfopin attenuates HDACi-mediated accumulation of H3K27ac on genomic regions associated with oncogenic pathways

We next examined epigenetic features that may underlie the inhibition of oncogenic pathways by the combined treatment (keeping the same experimental setup as shown in *Figure 2A*). Inhibition of HDACs is expected to increase and alter genome-wide histone acetylation patterns, including the H3K27ac mark associated with active promoters and enhancers (*Minucci and Pelicci, 2006*). Indeed, global quantification by single-molecule measurements of H3K27-acetylated nucleosomes, as well as western blot analysis, confirmed HDACi-mediated increase in H3K27ac global levels (*Figure 3A–C* and *Figure 3—figure supplement 1A*). Interestingly, treatment with Sulfopin alone had the opposite effect, reducing the percentage of acetylated nucleosomes (*Figure 3B and C* and *Figure 3—figure supplement 1A*). As acetylation is coupled with transcription activation, this is in line with the function of Sulfopin as an inhibitor of MYC transcriptional activity (*Dubiella et al., 2021*; *Poole and van Riggelen, 2017*). As a result, the global HDACi-mediated increase in histone acetylation was also attenuated in the combined treatment with Sulfopin (*Figure 3B and C* and *Figure 3—figure supplement 1A*).

To profile changes in the genomic distribution of H3K27ac and H3K27me3, we applied Cut&Run in SU-DIPG13 cells treated with Vorinostat alone or in combination with Sulfopin. We first examined the promoters of genes DE in the combined treatment (as clustered in *Figure 2G*). Genes comprising cluster 1 showed higher basal expression levels compared to genes from cluster 4, and these expression differences were also reflected in higher H3K27ac at these genes' promoters (*Figure 3—figure supplement 1B and C*). Moreover, in accordance with the reduction in RNA expression levels of genes comprising cluster 1, the combined treatment led to the strongest local reduction of H3K27ac levels associated with their promoters (*Figure 3D*). Specifically, we found that several of the oncogenes that comprise cluster 1, such as mTOR, and mTOR regulator SLC7A5, RELB, and COPB2, showed decreased levels of H3K27ac at their promoters following the combined treatment (*Kanai, 2022*; *Lee et al., 2013*; *Feng et al., 2021*; *Figure 3E* and *Figure 3—figure supplement 1D*). Cluster 1 genes also gained H3K27me3 on their promoters upon the combined treatment, supporting their epigenetic repression following Sulfopin and Vorinostat treatment (*Figure 3—figure supplement 1E*).

We further identified genomic regions in which histone acetylation is lost in a Sulfopin-dependent manner (i.e. H3K27ac peaks that are present in Vorinostat-treated cells, but are lost in the combined treatment with Sulfopin). Genome distribution analysis of these unique peaks revealed that the majority of them localize to distal regions (*Figure 3—figure supplement 1F*) and more than 60% of them localize to annotated enhancers. Importantly, genes associated with these enhancers were strongly enriched for oncogenic pathways such as RAS, MAPK, and PI3K-AKT signaling pathways (*Fernández-Medarde and Santos, 2011*; *Braicu et al., 2019*; *Rascio et al., 2021*; *Figure 3F* and *Figure 3—figure supplement 1G*). Specifically, the combined treatment reduced H3K27ac levels on several enhancers linked to genes involved in mTOR pathway and gliomagenesis, such as *AKT3*, *JAK1*, and *NFKB2* genes, that were shown to have critical role in malignant gliomas (*Peng et al., 2022*; *Mure et al., 2010*; *Ding et al., 2020*; *Tu et al., 2011*; *Raychaudhuri et al., 2007*), *SEC13* gene that indirectly activates mTOR (*Bar-Peled et al., 2013*; *Cai et al., 2016*), and *CD93*, a key regulator in glioma (*Langenkamp et al., 2015*; *Figure 3G* and *Figure 3—figure supplement 1H*). Concomitant with the lower H3K27ac levels associated with these enhancers, we also observed lower expression of the associated genes (*Figure 3H* and *Figure 3—figure supplement 1I*). Taken together, the results show attenuation of HDACi-mediated accumulation of H3K27ac levels upon combined treatment with Sulfopin, associated with promoters and enhancers of prominent oncogenes. This is in line with reduced expression of these oncogenic pathways, strengthening the therapeutic potential of the combined treatment.

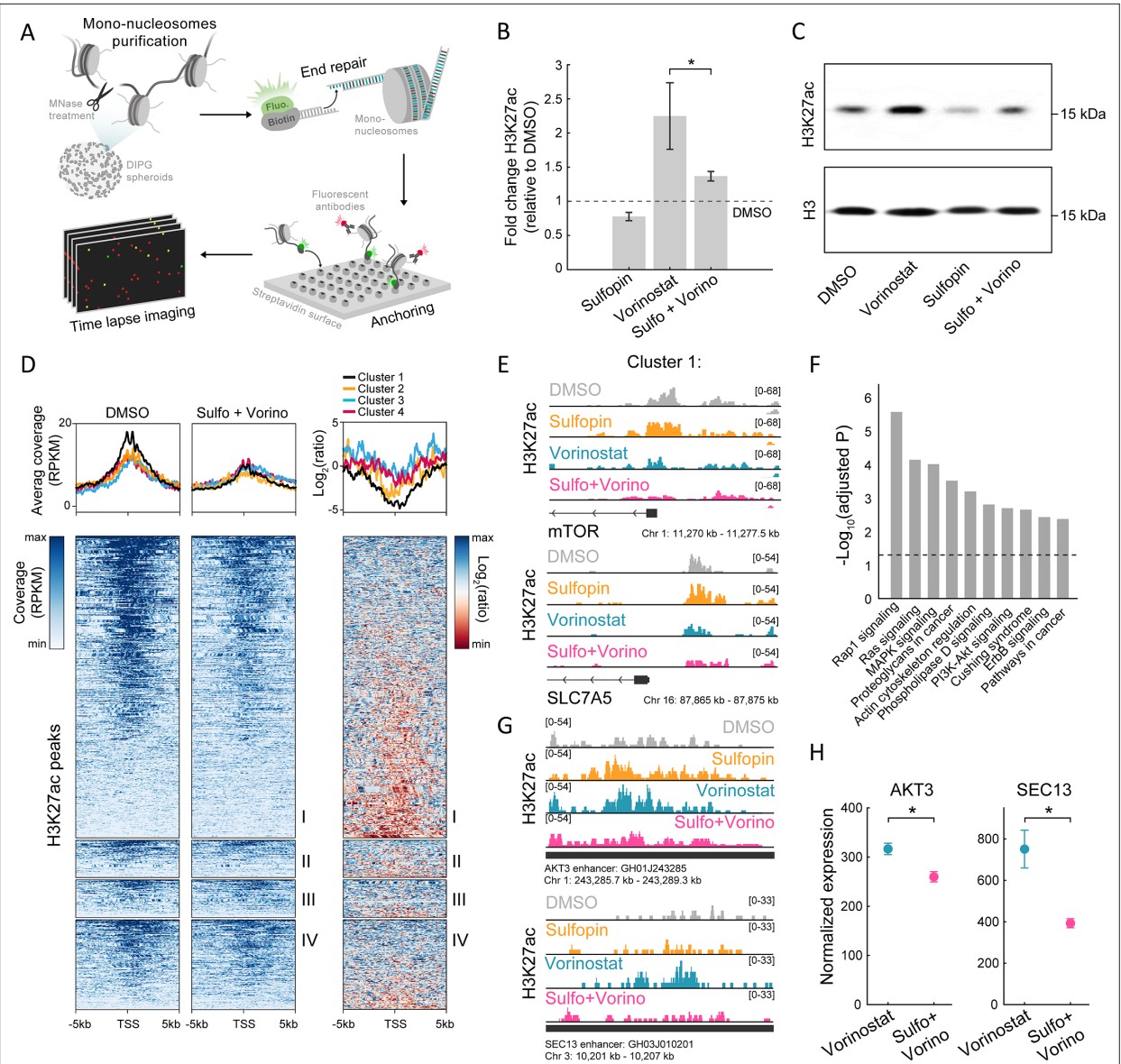

**Figure 3.** The combined treatment attenuates H3K27ac levels on oncogenic targets. (**A**) Scheme of the single-molecule imaging experimental setup (*Shema et al., 2016*): cell-derived mono-nucleosomes are anchored in a spatially distributed manner on polyethylene glycol (PEG)-coated surface. Captured nucleosomes are incubated with fluorescently labeled antibodies directed against the H3K27ac modification. Total internal reflection fluorescence (TIRF) microscopy is utilized to record the position and modification state of each nucleosome. Time series images are taken to allow detection of maximal binding events. (**B**) Single-molecule imaging quantification of the percentage of H3K27ac nucleosomes, in SU-DIPG13 cells treated with either Sulfopin (10 μM, 8 days), Vorinostat (1 μM, 72 hr), or the combination of Sulfopin and Vorinostat, normalized to DMSO. Mean fold±SE of at least two independent experiments is shown. H3K27ac global levels are lower in the combined treatment compared to cells treated solely with Vorinostat. *p<0.05 (two-sample t-test). (**C**) SU-DIPG13 cells were treated as in B, and analyzed by western blot using the indicated antibodies. (**D**) Left panel: Heatmap shows H3K27ac read coverage around the transcriptional start sites (TSS) (±5 kb) of the significantly differentially expressed (DE) genes shown in *Figure 2G*, in SU-DIPG13 cells treated with the combination of 10 μM Sulfopin and 1 μM Vorinostat versus DMSO. Average coverage is shown on top. Color intensity corresponds to the standardized expression. Clusters 1–4 are indicated. Right panel: The log2 ratio of H3K27ac read coverage in SU-DIPG13 cells treated with the combination of 10 μM Sulfopin and 1 μM Vorinostat vs. DMSO was calculated. Heatmap shows the ratio around the TSS (±5 Kb) of the significantly DE genes shown in *Figure 2G*, and average coverage is shown on top. Color intensity corresponds to the ratio between samples, low (red) to high (blue). Clusters 1–4 are indicated, with cluster 1 presenting the strongest local decrease in H3K27ac following the combined treatment compared to DMSO. (**E**) IGV tracks of *MTOR* and *SLC7A5* gene promoters, showing H3K27ac coverage in SU-DIPG13 cells treated as indicated. (**F**) Functional enrichment analysis of the genes linked to enhancers (top targets of high confident enhancers) marked with H3K27ac exclusively in SU-DIPG13 cells treated with Vorinostat, and not in the combined treatment. gProfiler algorithm (*Raudvere et al., 2019*) was used to calculate enrichment against the KEGG pathways DB (*Kanehisa and Goto, 2000*). Dashed line denotes adjusted p-value=0.05. Genes associated with

*Figure 3 continued on next page*

*Figure 3 continued*

Vorinostat-unique enhancers are enriched for oncogenic signaling pathways. (**G**) IGV track of *AKT3* and *SEC13* linked enhancers, showing H3K27ac coverage in SU-DIPG13 cells treated with 1 µM Vorinostat or the combination of 10 µM Sulfopin and 1 µM Vorinostat. (**H**) Normalized expression levels of *AKT3* and *SEC13* genes in SU-DIPG13 cells treated as in G. Mean ± SD of three technical repeats is shown. *p<0.05 (two-sample t-test).

The online version of this article includes the following source data and figure supplement(s) for figure 3:

**Source data 1.** Original scan for the western blot analysis shown in *Figure 3C*.

**Source data 2.** Annotation of the original scan for the western blot analysis shown in *Figure 3C*.

**Figure supplement 1.** H3K27ac levels decrease following the combined treatment specifically on oncogenes.

## The combined treatment reduces tumor growth in a DMG xenograft mouse model

Finally, we examined whether the combined inhibition of HDACs and PIN1 elicits an additive effect also in vivo. Thus, we established an H3-K27M DMG orthotopic xenograft model by injections of SU-DIPG13P* cells, engineered to express firefly luciferase, to the pons of immunodeficient mice (*Grasso et al., 2015*). Mice were given 10 days to develop tumors, and then randomly assigned to four groups: control mice treated with DMSO, mice treated with Sulfopin or Vorinostat as single agents, and mice treated with the combination of Sulfopin and Vorinostat. Both drugs were given daily and tumor growth was monitored by in vivo bioluminescence imaging. In line with our in vitro results, we found significant reduction in tumor growth following the combined treatment compared to control DMSO-treated mice (*Figure 4A and B*). Importantly, the reduction was also seen when comparing to mice treated solely with Vorinostat, supporting the additive value of Sulfopin to the well-known anti-tumorigenic effect of HDACi in H3-K27M gliomas (*Grasso et al., 2015*; *Figure 4A and B*). Of note, mice from all groups showed signs of severe dehydration and general deterioration after 18 days of treatment, and thus we could not measure a survival benefit for the combined treatment. This rapid deterioration is likely a result of the aggressiveness of the transplanted tumors and does not represent side effects of the treatment, as mice from all groups, including the non-treated mice, showed similar signs of deterioration.

To link our in vitro transcriptomic analysis to the in vivo effect on tumor growth, we assessed the percentage of H3-K27M cells that are positive to mTOR signal (double-positive), out of the total number of H3-K27M cells, by immunofluorescence staining. The combined treatment significantly reduced the fraction of the double-positive cells, compared to DMSO (*Figure 4C and D*). Of note, mTOR signal in the mouse cells surrounding the tumor showed no significant differences between the DMSO and the combination-treated mice (*Figure 4—figure supplement 1A and B*). Overall, these results verify the potency of this treatment in restricting oncogenic pathways also in vivo, and the potential clinical benefit of PIN1 and HDAC combined inhibition in H3-K27M-mutant DMGs.

## Discussion

Despite decades of efforts to improve treatment for children diagnosed with H3-K27M-mutant DMGs, the standard of care remains solely radiation, with no significant improvement in overall survival (*Johung and Monje, 2017*; *Hayden et al., 2021*; *Nagaraja et al., 2017*). High-throughput drug screens in patient-derived cells identified HDAC inhibition as a promising therapeutic strategy, and ongoing clinical trials revealed preliminary evidence of clinical benefit (*Lin et al., 2019*; *Cooney et al., 2018*) (NCT02717455, NCT03566199). However, emerging resistance to HDACi in glioma cells (*Grasso et al., 2015*), as well as limited therapeutic benefits of HDACi treatment as a sole agent (*Hennika et al., 2017*; *Leszczynska et al., 2021*), highlight the necessity to discover additional therapy modalities that could prolong survival and relieve symptoms.

In H3-K27M-mutant DMGs, as well as in many other malignancies, high MYC activity is correlated with poor prognosis and therefore considered as an attractive therapeutic target (*Buczkowicz et al., 2014*; *Oster et al., 2002*). Nevertheless, direct targeting of MYC is highly challenging, functional, and structural wise. MYC signaling, while amplified in cancer cells, is also essential to normal cell function (*Pelengaris et al., 2002*). Moreover, as a transcription factor, MYC lacks traditionally druggable binding pockets, hindering the development of direct small molecule binders (*Wang et al., 2021*). To overcome this, in this study we exploited Sulfopin, a novel, potent, and selective PIN1 inhibitor

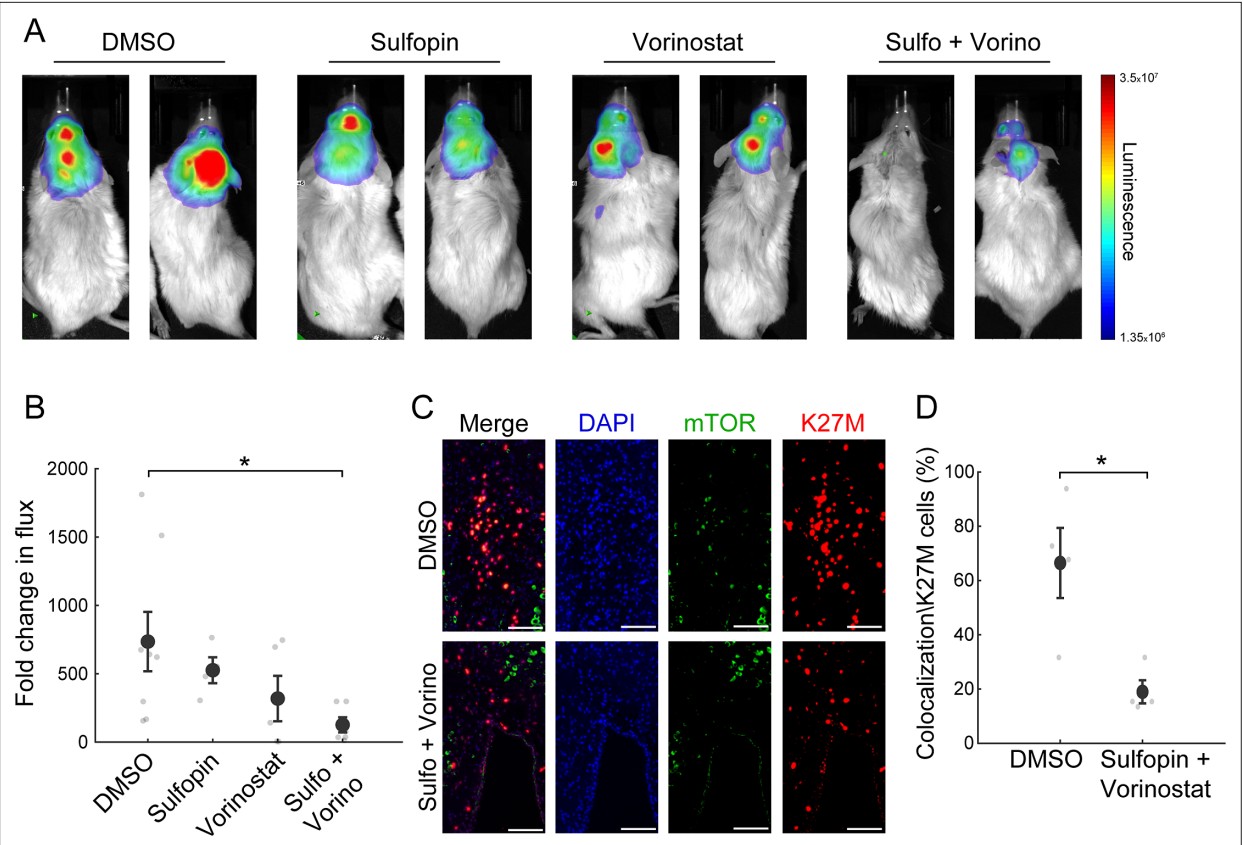

**Figure 4.** The combined treatment with Sulfopin and Vorinostat reduces tumor growth in vivo. (**A–B**) SU-DIPG13P* cells were injected to the pons of immunodeficient mice to form tumors. Ten days post injection, mice were treated for 18 days with either DMSO, Sulfopin, Vorinostat, or the combination of Sulfopin and Vorinostat. (**A**) In vivo bioluminescent imaging of diffuse midline glioma (DMG) xenografts following 18 days of treatment. The heatmap superimposed over the mouse head represents the degree of photon emission by DMG cells expressing firefly luciferase. (**B**) DMG xenograft tumor growth as measured by change in bioluminescent photon emission following 15 days of treatment with either DMSO (n=8), Sulfopin (n=4), Vorinostat (n=5), or the combination of Sulfopin and Vorinostat (n=6). Data points represent the fold change in maximum photon flux between day 3 and day 18 under treatment for each mouse. Mean ± SE is shown. *p<0.05 (two-tailed Mann-Whitney U-test). (**C–D**) Immunofluorescent staining of brain sections from mice injected with SU-DIPG13 cells and treated with DMSO (n=4) or the combination of Sulfopin and Vorinostat (n=4). (**C**) Representative fluorescence images of H3-K27M (red) and mTOR (green). Scale-bar = 100mm. (**D**) Percentage of mTOR-positive cells out of the total H3-K27M-positive cells (n=13–198 cells per field of view [FOV]). H3-K27M-positive cells show lower levels of mTOR following the combined treatment compared to DMSO. Mean ± SE is shown. *p<0.05 (two-tailed t-test).

The online version of this article includes the following figure supplement(s) for figure 4:

**Figure supplement 1.** The combined treatment with Sulfopin and Vorinostat results in tumor growth reduction in vivo.

(*Dubiella et al., 2021*). Preclinical studies have shown in vivo activity and negligible toxicity of Sulfopin in murine models of PDAC and neuroblastoma (*Dubiella et al., 2021*). While PIN1 is not overexpressed in DMG tumors, we show that Sulfopin, although not inhibiting MYC directly, downregulates MYC target genes in patient-derived DMG cells and affects cell survival in an H3-K27M-dependent manner. These results are in line with the contribution of MYC signaling to H3-K27M-driven tumorigenesis (*Pajovic et al., 2020*; *Dubois et al., 2022*), underscoring the therapeutic potential of Sulfopin for these aggressive brain tumors. Notably, despite a significant reduction in tumor size in vivo, the combined treatment did not increase mice survival. This is perhaps due to the relatively large tumors already formed at the onset of treatment, leading to rapid deterioration of mice in all experimental groups. Thus, further optimization of the modeling system and therapeutic regime is needed.

We showed that combining Sulfopin treatment with the HDAC inhibitor Vorinostat reduced cell viability by 80% in several tumor-derived DMG lines originating from different patients. Importantly, the combination resulted in an additive effect not only on cell viability, but also on transcriptional programs. A key oncogenic pathway that was robustly downregulated in the combined treatment, in

vitro and in vivo, is the mTOR signaling pathway. Preclinical and clinical data support a role for mTOR in gliomagenesis (*Akhavan et al., 2010*). Specifically, AKT gain or PTEN loss is detected in approximately 70% of DMG tumors, and a recent study showed high expression of mTOR in DMGs. Thus, the PTEN/AKT/mTOR pathway is suggested to play a key role in this disease (*Warren et al., 2012*; *Panditharatna et al., 2015*; *Zarghooni et al., 2010*; *Tsoli et al., 2018*). Attempts to restrict mTOR signaling using first-generation mTOR inhibitors primarily inhibited mTORC1, which often caused upregulation of mTORC2 (*Guertin and Sabatini, 2009*). The combined treatment with Sulfopin and Vorinostat led to robust inhibition of several target proteins in the mTOR signaling pathway, likely via alternative pathways. One potential candidate is the mTOR-dependent cell growth through MYC regulation, which may be impaired by Sulfopin treatment, inhibiting MYC transcriptional activity (*Zoncu et al., 2011*; *Schmidt et al., 2009*; *Sodi et al., 2015*; *Kuo et al., 2015*). Of note, drug agents targeting both mTORC1 and mTORC2 were shown to be highly effective in DMG cells as well as in murine models (*Miyahara et al., 2017*; *Flannery et al., 2018*).

Expression of the H3-K27M oncohistone is associated with a prominent increase in H3K27ac levels (*Lewis et al., 2013*; *Krug et al., 2019*; *Piunti et al., 2017*; *Furth et al., 2022*). Among the loci that accumulate this modification are repetitive elements and endogenous retroviruses, resulting in their aberrant expression, potentially inducing innate immune responses (*Krug et al., 2019*). However, H3-K27M cells do not show an increase in interferon stimulated genes (ISGs), preassembly due to high levels of MYC, which is potent antagonist of interferon responses (*Krug et al., 2019*; *Schlee et al., 2007*). Thus, dual targeting of both HDAC and MYC pathways has the potential of inducing robust expression of ISGs, thus promoting tumor cell death mediated by viral mimicry (*Jones et al., 2019*).

The high levels of H3K27ac in H3-K27M cells was also linked to activation of super-enhancers, which regulate the expression of oncogenes and genes that are associated with an undifferentiated state (*Nagaraja et al., 2017*). Here, we show that H3-K27M cells are vulnerable to transcriptional disruption mediated by the combined treatment with Sulfopin and Vorinostat. While treating cells with Vorinostat alone increased H3K27ac levels, as expected, the combined treatment attenuated the accumulation of this mark on promoters and enhancers of prominent oncogenes, resulting in their decreased expression. This Sulfopin-mediated attenuation could potentially result from downregulation of MYC targets, as it was recently shown that MYC overexpression results in H3K27ac gain on super-enhancers (*See et al., 2022*).

In summary, we propose a novel therapeutic strategy targeting epigenetic and transcriptional pathways in H3-K27M-mutant DMG tumors. HDAC inhibition, in combination with the novel MYC inhibitor, inhibited oncogenic transcriptional signatures and reduced tumor growth in vivo. Our findings emphasize epigenetic dependencies of H3-K27M-driven gliomas and highlight the therapeutic potential of combined treatments for this aggressive malignancy.

## Methods
### Cell cultures
All cell lines were maintained at 37°C with 5% $CO_2$. Exclusion of *Mycoplasma* contamination was monitored and conducted by test with EZ-PCR kit (Biological Industries #20-700-20).

### Glioma cultures
DIPG-derived cells, SU-DIPG13 (H3.3-K27M, female), SU-DIPG6 (H3.3-K27M, female), SU-DIPG17 (H3.3-K27M, male), SU-DIPG25 (H3.3-K27M, female), SU-DIPG50 (H3.3-K27M, sex unknown), SU-DIPG36 (H3.1-K27M, female), SU-DIPG38 (H3.1-K27M, female), SU-DIPG21 (H3.1-K27M, male), and SU-DIPG13P* were generated in the lab of Dr. Michelle Monje, Stanford University (*Grasso et al., 2015*). SU-DIPG13P* are a subclone of the patient-derived SU-DIPG13 pons culture that demonstrates more aggressive growth in vivo (*Lin et al., 2019*; *Nagaraja et al., 2017*).

K27M-KO SU-DIPG13, BT245-KO (sex information is unavailable), and corresponding control clones were generated in the lab of Prof. Nada Jabado, McGill University, as previously described by *Krug et al., 2019*.

Cells were cultured in Tumor Stem Media consisting of a 1:1 mixture of DMEM/F12 (Invitrogen, 31330038) and Neurobasal-A (Invitrogen, 10888022) media, with 1:100 addition of HEPES Buffer Solution (Invitrogen, 15630-080), MEM Sodium Pyruvate Solution (Invitrogen, 11360-070), MEM

Non-Essential Amino Acids Solution (Invitrogen, 11140-050), GlutaMAX-I (Invitrogen, 35050-061), and Antibiotic-Antimycotic (Invitrogen, 15240-096). The following additives were added freshly: B27-A (1:50, Invitrogen, 12587010), human FGF (20 ng/ml, Shenandoah Biotechnology, 100-146), human EGF (20 ng/ml, Shenandoah Biotechnology, 100-26), human PDGF-AA (20 ng/ml, Shenandoah Biotechnology, 100-16), human PDGF-BB (20 ng/ml, 100-18, Shenandoah Biotechnology), and heparin (10 ng/ml, STEMCELL Technologies, 07980). Doubling times were computed using: http://www.doubling-time.com/compute.php.

## Astrocytic-like cells culture

CRL-1718 cells, an astrocytic-like cells isolated from the brain of patient with astrocytoma, were kindly provided by Dr. D Michaelson, Tel-Aviv University, Israel. Cells were cultured in RPMI media (Biological Industries, #01-101-1A) supplemented with 10% heat-inactivated fetal bovine serum, 1 mM L-glutamine and 1% penicillin/streptomycin solution (all from Biological Industries).

## Isolation of total RNA, reverse transcription, and RT-qPCR

RNA was isolated from 2 million cells using the NucleoSpin kit (Macherey Nagel, #740955.50). 1 μg of each RNA sample was reverse-transcribed using Moloney Murine Leukemia Virus reverse transcriptase (M-MLV-RT, Promega, #M1701) and random hexamer primers (Thermo Scientific, #SO142). Real-time qPCR was performed using KAPA SYBR FAST mix (Kapa Biosystems, #KK4660) with a StepOne real-time PCR instrument (Applied Biosystems). For each gene, standard curve was determined and the relative quantity was normalized to GAPDH or HPRT mRNA (see Resources table in *Supplementary file 2*).

## Isolation of total RNA, bulk MARS-seq library preparation, and sequencing

RNA was isolated from SU-DIPG cells treated as indicated in *Supplementary file 3a*, using the NucleoSpin kit (Macherey Nagel, 740955). A bulk adaptation of the MARS-seq protocol (*Jaitin et al., 2014*; *Keren-Shaul et al., 2019*) was used to generate RNA-seq libraries for expression profiling. Briefly, 30 ng of input RNA from each sample was barcoded during reverse transcription and pooled. Following Agencourt Ampure XP beads cleanup (Beckman Coulter, A63880), the pooled samples underwent second strand synthesis and were linearly amplified by T7 in vitro transcription. The resulting RNA was fragmented and converted into a sequencing-ready library by tagging the samples with Illumina sequences during ligation, RT, and PCR. Libraries were quantified by Qubit (Thermo Fisher Scientific) and TapeStation (Agilent) as well as by qPCR for GAPDH housekeeping gene as described below. Sequencing was done on a NextSeq 75 cycles high-output kit (Illumina, 20024906).

## Bulk MARS-seq analysis

A median of 10.5–20 million reads were obtained for each dataset (*Supplementary file 3a*). MARS-seq analysis was preformed using the UTAP transcriptome analysis pipeline (*Kohen et al., 2019*). Reads were trimmed to remove adapters and low-quality bases using cutadapt (*Martin, 2011*) and mapped to the human genome (hg38, UCSC) using STAR v2.4.2a18 (*Dobin et al., 2013*) (parameters: –alignEndsType EndToEnd, –outFilterMismatchNoverLmax 0.05, –twopassMode Basic, –alignSoftClipAtReferenceEnds No). In short, the pipeline quantifies the 3′ of annotated genes (the 3′ region contains 1000 bases upstream of the 3′ end and 100 bases downstream). Counting was done using HTSeq-count in union mode (*Anders et al., 2015*). Genes having a minimum 5 UMI-corrected reads in at least one sample were considered. Normalization of the counts and differential expression analysis was done using DESeq2 (*Love et al., 2014*) (parameters: betaPrior = True, cooksCutoff = FALSE, independentFiltering = FALSE). Significantly DE genes were defined having adjusted p-value (with Benjamini and Hochberg procedure)≤0.05, |log2FoldChange|≥1, and baseMean ≥ 5. Heatmaps of gene expression were calculated using the log-normalized expression values (rld), with row standardization (scaling the mean of a row to 0, with standard deviation of 1). Clustering of gene expression (rld values) was done using Euclidean average linkage method, and visualized using Partek Genomics Suite 7.0 software (Partek, an Illumina company (2020). Partek Genomics Suite (Version 7.0)). Boxplot was used to visualize gene expression values (rld) of each DE gene from clusters 1 and 4 (in each of the three replicates).

## Expression analysis from published datasets

Gene expression data and histone mutation status of pediatric high-grade gliomas (*Mackay et al., 2017*) was downloaded from PedcBioPortal (https://pedcbioportal.kidsfirstdrc.org/). For matched tumor-normal samples, RPKM values were obtained from *Berlow et al., 2018* (S2_Table). For boxplots the log2 (RPKM+1) or z-score values were used (ggpubr R package). Differences between groups were assessed using two-sided 't_test' (rstatix R package). For heatmaps, gene expression values were obtained for genes from Molecular Signatures Database (MsigDB) signatures 'HALLMARK_MYC_TARGETS_V1', with MYC and PIN1 genes added manually. Heatmaps of gene expression were calculated using RPKM values, with row standardization (scaling the mean of a row to 0, with standard deviation of 1).

## Functional enrichment analysis

GSEA was performed as described in https://www.broadinstitute.org/software/gsea; *Subramanian et al., 2005*; *Mootha et al., 2003*. Genes detected by MARS-seq were pre-ranked according to their fold change (log2) in Sulfopin+Vorinostat vs. DMSO-treated cells. Pre-ranked GSEA was applied using the Human MSigDB Collections H: hallmark gene sets (*Liberzon et al., 2015*) and the C6: oncogenic signature gene sets (*Wiederschain et al., 2007*; *van Staveren et al., 2006*; *Wei et al., 2006*). Significantly downregulated gene sets were selected using cutoff of FDR q-value≤0.1.

Enrichr algorithm (*Kuleshov et al., 2016*) was used to compare significantly downregulated genes detected in the different treatments, against one or more of the following databases: the MSigDB hallmark (*Liberzon et al., 2015*), ChEA (*Keenan et al., 2019*), or ENCODE TF ChIP (*Dunham et al., 2012*).

gProfiler algorithm (*Raudvere et al., 2019*) was used to analyze genes that are putative targets of elite enhancers which overlap with Vorinostat-unique H3K27ac peaks (see Cut&Run analysis), against the KEGG (*Kanehisa, 2019*; *Kanehisa and Goto, 2000*) or Wiki (*Martens et al., 2021*) pathway databases.

## Cut&Run pulldown assay followed by high-throughput sequencing

Cut&Run assay was done as described in *Skene and Henikoff, 2017*; *Meers et al., 2019*, with slight modifications as follows: Cells were harvested and counted, with 200,000 cells taken per reaction. Permeabilized cells, bound to Concanavalin A-coated beads (Bangs Laboratories, BP531), were mixed with individual primary antibody (*Supplementary file 2*) and incubated overnight at 4°C while rotating. Secondary antibody, anti-rabbit HRP, was used as a negative control. pAG-MNase enzyme (generated in the Department of Life Sciences Core Facilities, WIS, using Addgene plasmid 123461) was added to each sample followed by incubation step of 1 hr at 4°C. Targeted digestion was done by 15 min incubation on ice block (0°C) under low salt conditions. DNA purification was done using Nucleospin gel and PCR clean-up kit (Machery-Nagel, 740609).

Libraries were prepared from 1 to 20 ng of DNA as previously described in *Blecher-Gonen et al., 2013*. Briefly, DNA fragments were repaired by T4 DNA polymerase (NEB, M0203), and T4 poly-nucleotide kinase (T4 PNK, NEB, M0201) was used to add a phosphate group at the 5' ends. An adenosine base was then added by Klenow fragments (NEB, M0212) to allow efficient ligation, using T4 quick ligase (NEB, M2200), of sequencing adapters, which contain a T-overhang. DNA fragments were amplified by PCR (Pfu Ultra II fusion, Agilent Technologies, 600670), which also introduced the Illumina-P5 adapter at one end of the molecule. SPRI beads were used to purify proper sized DNA fragments after each enzymatic step. Libraries were quantified by Qubit (Thermo Fisher Scientific) and proper fragment range (200–400 bp) was verified by TapeStation (Agilent). Sequencing was done on a NextSeq 500 instrument (Illumina, #20024904) using a V2 150 cycles mid output kit (paired-end sequencing).

Paired-end reads of each sample (~7.5 million median reads per sample, *Supplementary file 3b*) were preprocessed with cutadapt (*Martin, 2011*), to remove adapters and low-quality bases (parameters: --times 2 -q 30 -m 20), following evaluation of quality with FastQC. Reads were mapped to human genome (hg38, UCSC) using Bowtie version 2.3.5.1 (*Langmead and Salzberg, 2012*) (--local --very-sensitive-local --no-unal --no-mixed --no-discordant --dovetail -I 10 -X 700). Nucleosome fragments at the length >120 bp were selected from the remaining unique reads using picard-tools, and broad peaks were called using MACS2 against the HRP samples as background

control (parameters: -f BAMPE --SPMR --nomodel --extsize 100 --keep-dup auto -q 0.05). Analysis of genomic features was done using ChIPseeker (*Yu et al., 2015*). Reads coverage on TSS, gene body regions, specific gene-signatures, and CGI were visualized using ngs.plot (*Shen et al., 2014*). Annotation of CGI were downloaded for UCSC table browser. Association of peaks with specific genes was done using GREAT (*McLean et al., 2010*). Bigwig files were constructed from BAM alignments using deepTools2 suite (*Ramírez et al., 2016*), by 'bamCoverage' command, using RPKM normalization in 10 bp bins. Heatmaps and profiles were constructed in 'scale-regions' mode around peak summits, with missingDataAsZero parameter using 'plotHeatmap' and 'plotProfile' commands. In the heatmap of H3K27ac ratio (*Figure 3D*) coverage was calculated as the log2 between treatment and DMSO, on TSS of genes which were depicted as DE by MARS-seq. RPKM-normalized coverage files were visualized on the genome using IGV (2.8.6). H3K27ac peaks were further distinguished into Vorinostat-unique peaks (i.e. H3K27ac peaks that are present in Vorinostat-treated cells, but are lost in the combined treatment with Sulfopin), and to shared peaks (i.e. H3K27ac peaks that are present both in Vorinostat and in Sulfopin+Vorinostat-treated cells, with a minimum of 1 bp overlap) using bedtools (*Quinlan and Hall, 2010*). H3K27me3 peaks unique to Sulfopin treatment were defined similarly. To detect the overlap between the Vorinostat-unique peaks with enhancer regions, we used the GeneHancer database v5.9 (*Fishilevich et al., 2017*). We required a 30% overlap of the peak length with a known enhancer categorized as 'Elite' group, and excluded enhancers annotated to promoters regions.

## Cell viability assay by CellTiterGlo

For Sulfopin and Vorinostat combination experiments, 384-well plates were pre-plated with twofold dilution matrix of Vorinostat (MCE Cat#HY-10221) and Sulfopin (generous gift from Dr. Nir London's Lab, Weizmann Institute of Science), in five concentrations ranging from 120 to 10,000 nM (Vorinostat) and 2500 to 40,000 nM (Sulfopin). 1400 DIPG cells/well were plated into each well containing 75 µl of growth media, and subjected to drug treatment for 3 days (Vorinostat) or 8 days with pulse at day 4 (Sulfopin). Single-agent titrations were run in parallel. Following the treatment, cells were subjected to cell viability assay using CellTiterGlo assay (#G7572, Promega, USA) in accordance with the manufacturer's protocol. Luminescence signal was detected by luminescence module of PheraStar FS plate reader (BMG Labtech, Ortenberg, Germany). Viability data was analyzed using GeneData 15 (Switzerland).

For generating Sulfopin dosage curves in DIPG versus astrocyte cells, the appropriated amounts of Sulfopin or DMSO (control) were transferred using Echo Liquid Handler 550 (Beckman, USA) into white/white 384-well TC plate (Greiner, #781080). 1400 cells/well were dispensed by MultiDrop 384 dispenser (Thermo Scientific, USA) into 384-well plates. After 4 days, cells were subjected to cell viability assay using CellTiterGlo assay (#G7572, Promega, USA) in accordance with the manufacturer's protocol. Luminescence signal was detected by luminescence module of PheraStar FS plate reader (BMG Labtech, Ortenberg, Germany). Viability data was analyzed using GeneData 15 (Switzerland).

For all the other viability assays, cells were plated at a density of 2500 cells/well in 96-well plates and subjected to drug treatment for 3 days (Vorinostat-MCE Cat#HY-10221, GSK-J4-Selleck Cat#S7070, EPZ-6438-Selleck Cat#S7128) or 8 days (Sulfopin, generous gift from Dr. Nir London's Lab, and MM-102-Selleck Cat#S7265) (*Supplementary file 2*). At least two replicates were measured for each treatment. When treated for 8 days drug was also added at day 4. Cell viability was measured by CellTiterGlo assay (G7571, Promega) according to the manufacturer's instructions. Luminescence was measured by Cytation 5 plate reader and viability was compared to DMSO-treated control cells.

At least two technical replicates were averaged in each experiment, and mean±SE of all biological repeats was calculated. Two-sample t-test on all technical replicates was used to determine significant difference between the treatments. Bonferroni correction was applied whenever more than two comparisons were made.

## IC50 quantification

Sulfopin IC50 values were determined by 4-parameter log-logistic dose-response model using SynergyFinder 2.0 (*Ianevski et al., 2020*).

## BLISS index calculation

The BLISS index was calculated as the ratio between observed effect of combined drug treatment and expected effect (based on the calculated effect of combining single-drug treatment) (*Yadav et al., 2015*; *Bliss, 1939*):

$$\text{Bliss index} = \text{Observed/Expected} = \frac{J_{AB}}{(J_A \times J_B)}$$

$J_A$ is the effect on growth of drug A only compared to DMSO
$J_B$ is the effect on growth of drug B only compared to DMSO
$J_{AB}$ is the effect on growth of drug A and B together compared to DMSO
Synergy: Bliss < 1
Additive: Bliss = 1
Antagonist: Bliss > 1

## Western blot analysis

An equal number of cells from different treatments were suspended in Laemmli sample buffer (Bio-Rad, #1610747) containing 50 mM DTT (Promega, #V3151), vortexed and heated up to 98°C. Samples were loaded onto Novex WedgeWell 4–20%, Tris-Glycine 4–20% gel (Thermo Fisher Scientific, XP04205BOX) and electrophoresis was carried out in TG-SDS buffer (Bio-Rad #1610732) for 1–1.5 hr at 120 V. After electrophoresis, proteins were transferred to nitrocellulose membranes using transfer kit (Bio-Rad, #1704156), followed by brief rinse in TBS (Bio-Rad, #1706435) and blocking with 5% (wt/vol) milk powder in TBST (Tween20-Sigma, #P1379) for 60 min. The membranes were then rinsed in TBST and incubated with primary antibodies overnight at 4°C with gentle rocking. Primary antibodies (*Supplementary file 2*) were diluted according to the manufacturer's instructions in TBST containing 5% (wt/vol) milk powder. The following day the membranes were rinsed in TBST, then incubated with HRP-conjugated secondary antibody (*Supplementary file 2*) diluted according to the manufacturer's instructions in a solution containing 5% (wt/vol) milk powder in TBST. The membranes were then rinsed in TBST and dipped in an ECL WB Detection Reagent (Bio-Rad #1705061) prior to exposure. Membranes were imaged using a Bio-Rad ChemiDoc MP imaging system.

## Nucleosome preparation for single-molecule imaging

Nucleosomes for single-molecule imaging were extracted and labeled as described in *Furth et al., 2022*. Briefly, 2–2.5 million cells were collected, washed once with PBS supplemented with protease inhibitors cocktail, 1:100, Sigma P8340), and HDAC inhibitors (20 mM sodium butyrate, Sigma 303410, and 0.1 mM Vorinostat V-8477), and then resuspended in 0.05% IGEPAL (Sigma I8896) diluted in PBS (supplemented with inhibitors). Cell pellets were then lysed in parallel to chromatin digestion (100 mM Tris-HCl pH 7.5, 300 mM NaCl, 2% Triton X-100, 0.2% sodium deoxycholate, 10 mM CaCl$_2$) supplemented with inhibitors and Micrococcal Nuclease (Thermo Fisher Scientific, 88216). The suspension was incubated at 37°C for 10 min and the MNase reaction was inactivated by addition of EGTA at a final concentration of 20 mM. Lysate was cleared by centrifugation and nucleosomes were concentrated using an Amicon ultra-4 (Millipore, UFC810024). Inhibitors were supplemented following concentration. Nucleosomes labeling the following reaction was used: NEBuffer 2 (NEB B7202), inhibitors (as detailed above, 0.25 mM MnCl$_2$, 33 µM fluorescently labeled dATP (Jena Bioscience, NU-1611-Cy3), 33 µM biotinylated dUTP (Jena Bioscience, NU-803-BIOX), 1.5 µl of Klenow Fragment (3'→5'exo-, NEB, M0212S), and 1.5 µl of T4 Polynucleotide Kinase (NEB, M0201L). Samples were incubated at 37°C for 1 hr and then inactivated by the addition of EDTA at a final concentration of 20 mM. Nucleosomes were then purified on Performa Spin Columns (EdgeBio, 13266) followed by the addition of inhibitors.

## Surface preparation for single-molecule imaging

PEG-biotin microscope slides were prepared as described in *Furth et al., 2022*. Ibidi glass coverslips (25×75 mm$^2$, IBIDI, IBD-10812) were cleaned with (1) ddH$_2$O (3× washes, 5 min sonication, 3× washes), (2) 2% Alconox (Sigma 242985, 20 min sonication followed by 5× washes with ddH$_2$O), (3) 100% acetone (20 min sonication followed by 3× washes with ddH$_2$O). Slides were then incubated in 1 M KOH solution for 30 min while sonicated (Sigma 484016), followed by 3× washes with ddH$_2$O.

Slides were sonicated for 10 min in 100% HPLC EtOH (J.T baker 8462-25) and then incubated for 24 min in a mixture of 3% 3-aminopropyltriethoxysilane (ACROS Organics, 430941000) and 5% acetic acid in HPLC EtOH, with 1 min sonication in the middle. Following washes with HPLC EtOH (3×) and ddH$_2$O (3×) slides were dried and mPEG:biotin-PEG solution was applied (20 mg Biotin-PEG [Laysan, Biotin-PEG-SVA-5000] and 180 mg mPEG [Laysan, MPEG-SVA-5000] dissolved in 1560 μl 0.1 M sodium bicarbonate [Sigma, S6297] on one surface followed by the assembly of another surface on top). Surfaces were incubated overnight in a dark humid environment and at the following day, were washed with ddH$_2$O and dried. MS (PEG) 4 (Thermo Fisher Scientific, TS-22341) was diluted in 0.1 M of sodium bicarbonate to a final concentration of 11.7 mg/ml and applied on the surfaces for overnight incubation in dark humid environment. Surfaces were washed with ddH$_2$O and dried.

## Single-molecule imaging of histone modifications

PEG-biotin-coated coverslips were assembled into Ibidi flowcell (Sticky Slide VI hydrophobic, IBIDI, IBD-80608). Streptavidin (Sigma, S4762) was added to a final concentration of 0.2 mg/ml followed by an incubation of 10 min. TetraSpeck beads (used for image alignment, Thermo Fisher Scientific, T7279) diluted in PBS were added and incubated on surface for at least 30 min. Labeled nucleosomes were incubated for 10 min in imaging buffer (10 mM MES pH 6.5 [Boston Bioproducts Inc, NC9904354], 60 mM KCl, 0.32 mM EDTA, 3 mM MgCl$_2$, 10% glycerol, 0.1 mg/ml BSA [Sigma, A7906], 0.02% Igepal [Sigma, I8896]) to allow immobilization via biotin-streptavidin interactions and washed with imaging buffer. H3K27ac antibody was diluted in imaging buffer (1:2000) and incubated for 30 min. All positions (80–100 fields of view [FOVs] per experiment) were then imaged by a total internal reflection fluorescence (TIRF) microscope by Nikon (Ti2 LU-N4 TIRF) every 10 min (10–15 cycles).

Image analysis was performed with Cell Profiler image analysis tools (http://www.cellprofiler.org/) as described in *Furth et al., 2022*. Image analysis is done in two steps: (1) Time-lapse images of antibody binding events and TetraSpeck beads are aligned, stacked, and summed to one image. Antibody spots and TetraSpeck beads spots were distinguished based on the size of the spot. (2) Stacked images are aligned to the initial images of the nucleosomes based on TetraSpeck beads location spots, and only binding events that align with nucleosomes are filtered and saved for further analysis. To evaluate random co-localization (negative control), each stacked image is aligned to a 90° flipped image of the initial nucleosomes. The use of TetraSpeck beads and a highly accurate microscope stage allowed alignment of images with shifts of up to 30 pixels. Nucleosomes are initially distributed on the surface in low density to minimize overlap between spots. Average percentage of modified nucleosomes over all FOV images was calculated in each experiment. Mean±SE of three to five biological repeats was calculated. Two-sample t-test on all biological repeats was used to determine significant difference between the treatments.

## Mouse injection and treatment

All animal studies were approved by the Weizmann Institutional Board and were performed in accordance with the Israeli law and the guidelines of the Institutional Animal Care and Use Committee (IACUC) approval number 07821021-1 and IRB approval number 1528-1. Mice were housed and handled in a pathogen-free, temperature-controlled (22C±1°C) facility on a 12 hr light/dark cycle. Animals were fed a regular chow diet and given ad libitum access to food and water. All surgeries were performed under isoflurane anesthesia, and every effort was made to minimize suffering.

Single-cell suspension of SU-DIPG13P* or SU-DIPG13 cells were stereotactically injected into the pons of 12-week-old male NGS mice (NOD-SCID-IL2R gamma chain-deficient, The Jackson Laboratory) as previously described in *Venkatesh et al., 2019*. Injection coordinates were: 0.8 mm posterior to lambda, 1 mm lateral to the sagittal suture, and 5 mm deep. Briefly, the skull of the mouse was exposed, and a small bore hole (0.5 mm) was made using a high-speed drill at the appropriate stereotactic coordinates. Approximately 300,000–400,000 cells in 3 μl volume were injected at a speed of 0.3 μl per minute into the pons with a 26-gauge Hamilton syringe; following 10 min pause, the needle was removed at a speed of 0.2 mm/min. After closing the scalp, mice were placed on a heating pad and returned to their cages after full recovery. All injected mice that recovered were randomized to treatment groups, and treatment started 10 days (SU-DIPG13P*) or 5 weeks (SU-DIPG13) post injection. Treatment was given daily for 18 days, by intraperitoneal injection according to the following groups: Vorinostat 200 mg/kg daily (LC, V-8477-SU-DIPG13P*; MCE, HY-10221- SU-DIPG13), Sulfopin

40 mg/kg (gift from Dr. Nir London's Lab), combination 200 mg/kg Vorinostst and 40 mg/kg Sulfopin, control mice received solvent 15% DMSO in 50% PEG400 (Sigma) in 0.15 M NaCl.

## Tumor size imaging using IVIS

For bioluminescence imaging, mice received 100 µl D-luciferin (33 mg/ml; Perkin-elmer, Cat#122799) intraperitoneally 10 min before isoflurane anesthesia and were placed in LagoX Imaging System (Spectral Instruments Imaging).

The bioluminescence signal was measured using the region of interest (ROI) tool in LagoX software (Spectral Instruments Imaging). The fold change in total photon flux (p/s) over time was used to compare tumor growth between the groups. Mann-Whitney U-test (two-tailed) was used to determine significant difference between the DMSO and Sulfopin+Vorinostat-treated mice.

## Mouse brain slides staining

Brains were collected, fixed in 4% paraformaldehyde, and embedded in paraffin blocks. For H3-K27M and mTOR staining, slides were de-paraffinized (1 - Xylene-Bio-lab #242500; 2 - Xylene; 3 - Eth 100%-Bio-lab #05250501; 4 - Eth 96%; 5 - Eth 70%;) and incubated in acetone (Bio-lab, #010352) at –20°C. Slides were incubated in hydrogen peroxide blocking solution (200 ml Methanol-Bio-lab #136805, 1% HCl-Bio-lab #084102, 3% $H_2O_2$- JT Baker #2186-01) for blocking endogenous peroxidase activity. Antigen retrieval was performed using citric acid buffer at pH 6, followed by blocking with 20% NHS and 0.5% Triton in PBS. Rabbit anti-mTOR (1:100, *Supplementary file 2*) was diluted in 2% NHS and 0.5% Triton and was incubated overnight at room temperature. Slides were then incubated with HRP-conjugated goat anti-rabbit (1:100, *Supplementary file 2*) diluted 2% NHS for 1.5 hr followed by OPAL 650 (1:500, Akoya FP1496001KT) reagent incubation for 15 min. Antibodies were stripped using microwave treatment with citric acid at pH 6, blocked and again incubated with rabbit anti H3-K27M (1:100 *Supplementary file 2*) overnight at room temperature. Slides were then incubated with HRP-conjugated goat anti-rabbit (1:100, *Supplementary file 2*) diluted 2% NHS for 1.5 hr followed by OPAL 570 (1:500, Akoya FP1488001KT) reagent incubation for 15 min. Slides were then washed with PBS and incubated with Hoechst (1:1000) for 1 min, followed by mounting with 25×75×1 mm³ glass slides using Aqua-Poly/Mount Mounting Medium. All slides were scanned using the Panoramic MIDI slide scanner (3D Histech, Hungary) at ×20 magnification, to receive a full scan of the tumor section.

For H3-K27M and mTOR co-localization analysis, double stained cells were counted using ImageJ. First, threshold-based segmentation was used to detect positive cells in each of the channels and the image calculator feature was used to find the double positive cells. The total number of double-positive cells was then divided by the total number of H3-K27M-positive cells to calculate the percentage co-localization/H3-K27M cells. Similar-sized FOVs were used for all the images analyzed, and 13-198 H3-K27M-positive cells were detected per FOV (an average of 90 cells per FOV). For mTOR signal quantification in non-tumor cells (mice cells), an ROI was selected in a location on the section with no positive H3-K27M cells. Threshold-based segmentation was used to detect the percentage mTOR stained area.

## Statistical analysis

Unless noted otherwise, p-values were determined using two-sample t-test and are indicated in the figure legends.

Pearson correlation coefficient was used to assess the relationship between all variables. Scatter plot was generated using ggscatter (R ggpubr package).

## Acknowledgements

We thank H Keren-Shaul, D Robbins, Y Elazari, and N Adler (G-INCPM, WIS) for their help with NGS; N Morris for her help with conducting part of the cell viability assays; C Raanan and M Zerbib for their contribution in establishing the DIPG mouse model; I Savchenko for his help with immunohistochemistry; R Gabizon for generously providing Sulfopin for in vitro and in vivo experiments and helping with Sulfopin handling; S Fishilevich for providing the GeneHancers data; L Segev for computational framework for single-molecule image analysis; N Harpaz and O Beresh for helping with cultures maintenance; O Griess for helping with transcriptomic protocols; S Gillespie for providing valid information on DMG cells growth and maintenance; and I Ulitsky for providing the pAG-MNase enzyme. We

are thankful to O Golani for helping with the immunohistochemistry image analysis. We are grateful to M Monje for generously sharing with us the SU-DIPG13, SU-DIPG13P*, SU-DIPG6, SU-DIPG17, SU-DIPG25, SU-DIPG36, SU-DIPG38, SU-DIPG21, and SU-DIPG50 cells. We are grateful to N Jabado for her kind gift of the isogenic BT245 and SU-DIPG13 cultures. We would like to express our sincere gratitude to D Deitch for producing part of the graphs, designing the figures and illustrations, and helping with statistical analysis. ES is an incumbent of the Lisa and Jeffrey Aronin Family Career Development chair and is also supported by Henry Chanoch Krenter Institute for Biomedical Imaging and Genomics. This research was supported by grants from the European Research Council (ERC801655), The Israel Science Foundation (1881/19), Emerson Collective, and The Israel Cancer Research Fund: Research Career Development Award.

# Additional information

## Competing interests

Nir London: N.L. is an inventor on a patent describing Sulfopin (US 2021/0332024 A1). The other authors declare that no competing interests exist.

## Funding

| Funder | Grant reference number | Author |
| --- | --- | --- |
| European Research Council | ERC801655 | Efrat Shema |
| The Israel Science Foundation | 1881/19 | Efrat Shema |
| Emerson Collective | | Efrat Shema |
| The Israeli Cancer Research Fund | | Efrat Shema |

The funders had no role in study design, data collection and interpretation, or the decision to submit the work for publication.

## Author contributions

Danielle Algranati, designed the study, conducted the experiments, performed bioinformatics analysis, conducted the in-vivo image analysis and wrote the manuscript; Roni Oren, established the xenograft mice model and performed in-vivo experiments and conducted the in-vivo image analysis; Bareket Dassa, performed bioinformatics analysis; Liat Fellus-Alyagor, conducted the in-vivo image analysis; Alexander Plotnikov, conducted part of the viability assays; Haim Barr, conducted part of the viability assays; Alon Harmelin, established the xenograft mice model; Nir London, suggested and provided Sulfopin and valuable usage information; Guy Ron, conducted the in-vivo image analysis; Noa Furth, designed the study, performed bioinformatics analysis and wrote the manuscript; Efrat Shema, designed the study, performed bioinformatics analysis and wrote the manuscript

## Author ORCIDs

Roni Oren ⬚ https://orcid.org/0000-0003-1228-412X
Guy Ron ⬚ http://orcid.org/0000-0002-8129-7146
Noa Furth ⬚ https://orcid.org/0000-0001-8728-519X
Efrat Shema ⬚ https://orcid.org/0000-0002-3718-593X

## Ethics

All animal studies were approved by the Weizmann institutional board and were performed in accordance with the Israeli law and the guidelines of the Institutional Animal Care and Use Committee (IACUC) approval number 07821021-1 and IRB approval number 1528-1. Mice were housed and handled in a pathogen-free, temperature-controlled (22C ± 1C) facility on a 12 h light/dark cycle. Animals were fed a regular chow diet and given ad libitum access to food and water. All surgeries were performed under isoflurane anesthesia, and every effort was made to minimize suffering.

Reviewer #2 (Public Review): https://doi.org/10.7554/eLife.96257.3.sa1
Author response https://doi.org/10.7554/eLife.96257.3.sa2

## Additional files

### Supplementary files

• Supplementary file 1. List of differentially expressed genes and their corresponding cluster. Related to *Figure 2G*.

• Supplementary file 2. Reagents and resources.

• Supplementary file 3. Summary information of NGS libraries. (a) Summary information of MARS-seq libraries. (b) Summary information of Cut&Run libraries.

• MDAR checklist

### Data availability

All sequencing data is deposited in NCBI's Gene Expression Omnibus (GEO) and available through GEO series accession number GSE221614.

The following dataset was generated:

| Author(s) | Year | Dataset title | Dataset URL | Database and Identifier |
| --- | --- | --- | --- | --- |
| Algranati D, Oren R, Dassa B, Fellus-Alyagor L, Plotnikov A, Barr H, Harmelin A, London N, Ron G, Furth N, Shema E | 2024 | Dual Targeting of Histone Deacetylases and MYC as Potential Treatment Strategy for H3-K27M Pediatric Gliomas | https://www.ncbi.nlm.nih.gov/geo/query.acc.cgi?acc=GSE221614 | NCBI Gene Expression Omnibus, GSE221614 |

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
