## [Editor Report · eLife assessment]

This work contributes to the study of H3-K27M mutated pediatric gliomas. It **convincingly** demonstrates that the concomitant targeting of histone deacetylases (HDACs) and the transcription factor MYC results in a notable reduction in cell viability and tumor growth. This reduction is linked to the suppression of critical oncogenic pathways, particularly mTOR signaling, emphasizing the role of these pathways in the disease's pathogenesis. The current version of the manuscript is **important** because it unveils a vulnerability from dual targeting HDACs and MYC in the context of pediatric gliomas. This work will be of interest to cancer epigenetics and therapeutics research, with a focus on the neuro-oncology field.

---

## [Referee Report · Reviewer #2 (Public Review)]

This study by Algranati et al. is a important contribution to our understanding of H3-K27M pediatric gliomas. It convincingly demonstrates that the concomitant targeting of histone deacetylases (HDACs) and MYC, through a combination therapy of Sulfopin and Vorinostat, results in a notable reduction in cell viability and tumor growth. This reduction is linked to the suppression of critical oncogenic pathways, particularly mTOR signaling, emphasizing the role of these pathways in the disease's pathogenesis. The manuscript is a step forward in the field, as it unveils a vulnerability from dual targeting HDACs and MYC in the context of pediatric gliomas.

Comments on revised version

The authors have nicely explained their rationale for dose selection, treatment timing, and the relationship between MYC expression and sensitivity to the combined treatment. They have also clarified the experimental conditions for the in vitro and in vivo studies, ensuring consistency across the various analyses.

Overall, the authors have been responsive to the reviewers' comments and have made appropriate revisions to improve the clarity and robustness of their study.

---

## [Author Response]

The following is the authors’ response to the original reviews.

**Public Reviews:**

**Reviewer #1 (Public Review):**
Summary:This is an interesting study that utilizes a novel epigenome profiling technology (single molecule imaging) in order to demonstrate its utility as a readout of therapeutic response in multiple DIPG cell lines. Two different drugs were evaluated, singly and in combination. Sulfopin, an inhibitor of a component upstream of the MYC pathway, and Vorinostat, an HDAC inhibitor. Both drugs sensitized DIPG cells, but high (>10 micromolar) concentrations were needed to achieve half-maximal effects. The combination seemed to have some efficacy in vivo, but also produced debilitating side-effects that precluded the measurement of any survival benefit.

We thank the reviewer for deeply evaluating our work and acknowledging the use of multiple experimental strategies to explore the effect of combination therapy on DMG cells. Of note, all mice in our experiment experienced deterioration (including the control mice and those treated with single agents). Thus, it is not the combination of drugs that led to the debilitating side-effects; the mice deteriorated due to the extremely aggressive tumor cells, forming relatively large tumors prior to the treatment onset, calling for further optimization of the therapeutic regime.

We modified the text in the results section to clarify this point (lines 238-241): “This rapid deterioration is likely a result of the aggressiveness of the transplanted tumors and does not represent side effects of the treatment, as mice from all groups, including the non-treated mice, showed similar signs of deterioration”.

We also elaborate on this in the discussion (lines 272-276): “Notably, despite a significant reduction in tumor size in-vivo, the combined treatment did not increase mice survival. This is perhaps due to the relatively large tumors already formed at the onset of treatment, leading to rapid deterioration of mice in all experimental groups. Thus, further optimization of the modeling system and therapeutic regime is needed.” We truly hope that further studies will allow better assessment of this drug combination in various models.

Strengths:Interesting use of a novel epigenome profiling technology (single molecule imaging).Weaknesses:The use of this novel imaging technology ultimately makes up only a minor part of the study. The rest of the results, i.e. DIPG sensitivity to HDAC and MYC pathway inhibition, have already been demonstrated by others (Grasso Monje 2015; Pajovic Hawkins 2020, among others). The drugs have some interesting opposing effects at the level of the epigenome, demonstrated through CUT&RUN, but this is not unexpected in any way. The drugs evaluated here also didn't have higher efficacy, or efficacy at especially low concentrations, than inhibitors used in previous reports. The combination therapy attempted here also caused severe side effects in mice (dehydration/deterioration), such that an effect on survival could not be determined. I'm not sure this study advances knowledge of targeted therapy approaches in DIPGs, or if it iterates on previous findings to deliver new, or more efficient, mechanistic or therapeutic/pharmaclogic insights. It is a translational report evaluating two drugs singly and in combination, finding that although they sensitise cells in vitro, efficacy in vivo is limited at best, as this particular combination cannot progress to human translation.

We thank the reviewer for pointing out the strengths and weaknesses of our work. As far as we know, while many studies demonstrated upregulation of the MYC pathway in DIPG, this is the first study that shows inhibition of this pathway (via PIN1) as a therapeutic strategy. While it is clear from the literature that MYC inhibition may pose therapeutic benefit, the development of potent MYC inhibitors is highly challenging due to its structure and cellular localization. Of note, in the 2020 paper, Pajovic and colleagues inhibited MYC by transfecting the cells with a plasmid expressing a specific inhibitory MYC peptide (Omomyc); while this strategy works well for cell cultures, the clinical translation requires different delivery strategies. Sulfopin is a small molecule inhibitor that can be used *in-vivo* and potentially in clinical studies. Thus, we believe that our study offers a novel strategy, as well as mechanistic insights, regarding the potential use of Sulfopin and Vorinostat to treat DIPG.

As noted above, the combination therapy did not cause side effects, but rather the aggressiveness of the tumors. We did not notice specific toxicity in the mice treated with Sulfopin alone, or the combined treatment. Furthermore, Dubiella et al. extensively examined toxicity issues and did not observe adverse effects or weight loss when administrating Sulfopin in a dose of 40 mg kg–1.

Optimization of the model and treatment regime (# of cells injected, treatment starting point, etc.) may have allowed us to reveal survival benefits. Yet, these are highly complicated and expensive experiments; unfortunately, we did not have the resources to perform them within the scope of this revision. Importantly, within the current manuscript, we show the effect of this drug combination in reducing the growth of DMG cells *in-vitro* and *in-vivo*, laying the framework for follow-up exploration in future studies. Furthermore, the epigenetic and transcriptomic profiling shed light on the molecular mechanisms that drive these aggressive tumors.

**Reviewer #2 (Public Review):**
Summary:The study by Algranati et al. introduces an exciting and promising therapeutic approach for the treatment of H3-K27M pediatric gliomas, a particularly aggressive brain cancer predominantly affecting children. By exploring the dual targeting of histone deacetylases (HDACs) and MYC activation, the research presents a novel strategy that significantly reduces cell viability and tumor growth in patient-derived glioma cells and xenograft mouse models. This approach, supported by transcriptomic and epigenomic profiling, unveils the potential of combining Sulfopin and Vorinostat to downregulate oncogenic pathways, including the mTOR signaling pathway. While the study offers valuable insights, it would benefit from additional clarification on several points, such as the rationale behind the dosing decisions for the compounds tested, the specific contributions of MYC amplification and H3K27me3 alterations to the observed therapeutic effects, and the details of the treatment protocols employed in both in-vitro and in-vivo experiments.

We thank the reviewer for evaluating our work and recognizing its potential for the DMG research field. We address in detail below the important comments regarding the treatment protocols and dosing decisions.

Clarification is needed on how doses were selected for the compounds in Figure S2A and throughout the study. Understanding the basis for these choices is crucial for interpreting the results and their potential clinical relevance. IC50s are calculated for specific patient derived lines, but it is not clear how these are used for selecting the dose.

We thank the reviewer for these important comments. For the epigenetic drugs shown in Figure S2A, we followed published experimental setups; for EPZ6438, GSKJ4, Vorinostat and MM-102 we chose the treating concentrations according to Mohammad et al. 2017, Grasso et al. 2015 and Furth et al. 2022, accordingly. For Sulfopin, we conducted a dedicated dose curve analysis (shown in Figure 1E), indicating only a mild effect on viability and relatively high IC-50 values as a single agent. Since we aimed to test the ability of a combined treatment to additively reduce viability, we used a sub-IC50 concentration for Sulfopin in these experiments. We added this information in lines 123 and 131-132.

Finally, following the results obtained in the experiment shown in Figure S2A, we conducted a full dose-curve analysis of the combined treatment in multiple DMG patient-derived cells (figure 2B and S2C), to identify a combination of concentrations that provides an additive effect (as indicated by BLISS index in figure 2C and S2E). Of note, for downstream analysis of the molecular mechanisms underlying the treatment response (RNAseq and Cut&Run), we intentionally used concentrations that provide an additive BLISS index, but do not completely abolish the culture, to allow for cellular analysis (i.e. 10uM Sulfopin and 1uM Vorinostat).

The introduction mentions MYC amplification in high-grade gliomas. It would be beneficial if the authors could delineate whether the models used exhibit varying degrees of MYC amplification and how this factor, alongside differences in H3K27me3, contributes to the observed effects of the treatment.

The reviewer highlights an important part of our study relating to the MYC-dependent sensitivity of the proposed treatment combination. Since high expression of MYC can be mediated by different molecular mechanisms and not only genomic amplification, we directly quantified mRNA levels of MYC by qPCR (shown in figure S2G) in order to explore its relationship with cellular response to Sulfopin and Vorinostat. Indeed, cultures that express high levels of MYC mRNA were more sensitive to Sulfopin treatment alone (figure S1P) and to the combined treatment (figure 2D-E). We also relate to these findings in lines 103-106 and 142-147 of the results section. Importantly, in cultures that express high levels of MYC (SU-DIPG13 as an example), we see downregulation of MYC targets upon the combined treatment, supporting the notion that this treatment affects viability by attenuation of MYC signaling.

In Figure 2A, the authors outline an optimal treatment timing for their in vitro models, which appears to be used throughout the figure. It would be helpful to know how this treatment timing was selected and also why Sulfopin is dosed first (and twice) before the vorinostat. Was this optimized?

As PIN1 regulates the G2/M transition, its inhibition by Sulfopin delays cell cycle progression (Yeh et al. 2007). Thus, in order to observe a strong viability difference in culture, a prolonged treatment period of 8-9 days is required (Dubiella et al., 2021). To maintain an active concentration of the drug during this long time period, we added a Sulfopin pulse (2nd dose) to achieve a stronger effect on cell viability. We and others noticed that, unlike Sulfopin, the effect of Vorinostat on viability is rapid and can be clearly seen after 2-3 days of treatment. Thus, we added this drug only after the 2nd dose of Sulfopin. We now relate to the mode of action of Sulfopin in lines 79-81.

It should be clarified whether the dosing timeline for the combination drug experiments in Figure 3 aligns with that of Figure 2. This information is also important for interpreting the epigenetic and transcriptional profiling and the timing should be discussed if they are administered sequentially (also shown in Figure 2A).I have the same question for the mouse experiments in Figure 4.

The reviewer is correct that this information is critical for evaluating the results. In order to link the expression changes to the epigenetic changes, we kept the same experimental conditions in both the Cut&Run and RNA-seq experiments (shown in figures 2-3). We added this information to the text in line 184.

For the *in-vivo* studies of HDAC inhibition (Figure 4), we followed published protocols (Ehteda et al. 2021). In these experiments both drugs were administrated simultaneously every day. We added this information to the text in line 231-232. It may be that changing the admission regime may improve the efficacy of the drug combination, which remains to be tested in future studies.

The authors mention that the mice all had severe dehydration and deterioration after 18 days. It would be helpful to know if there were differences in the side effects for different treatment groups? I would expect the combination to be the most severe. This is important in considering the combination treatment.

As noted in our response to Reviewer #1, all mice in our experiment experienced deterioration (including the control mice and those treated with single agents- we could not observe any differences between the groups). This is due to the extremely aggressive tumor cells, forming relatively large tumors prior to the treatment onset, calling for further optimization of the system and therapeutic regime (# of cell injected, treatment starting point, etc.). Unfortunately, this model is very challenging (especially the injection of cells to the pons of the mice brains, which requires unique expertise and is associated with mortality of some of the mice). Thus, these are highly complicated and expensive experiments; unfortunately, we did not have the resources to repeat and optimize the treatment protocol within the scope of this revision. Of note, Dubiella et al. extensively examined toxicity issues and did not observe adverse effects or weight loss when administrating Sulfopin in a dose of 40 mg kg–1. In our model, the side effects were caused by the tumors rather than the drugs.

Minor Points:(1) For Figure 1F, reorganizing the bars to directly compare the K27M and KO cell lines at each dose would improve readability of this figure.

We have changed figure 1F as the reviewer suggested.

(2) In Figure 4D, it would be helpful to know how many cells were included (or a minimum included) to calculate the percentages.

We added the number of H3-K27M positive cells detected per FOV to the figure legend and method section (n=13-198 cells per FOV). Of note, while we analyzed similar-sized FOVs, the number of tumor cells varied between the groups, with the treated group presenting a lower number of H3-K27M cells (due to the effect of the treatment on tumor growth). To account for this difference, we calculated the portion of mTOR-positive cells out of the tumor cells.

**Reviewer #3 (Public Review):**
Summary:The authors use in vitro grown cells and mouse xenografts to show that a combination of drugs, Sulfopin and Vorinostat, can impact the growth of cells derived from Diffuse midline gliomas, in particular the ones carrying the H3 K27M-mutations (clinically classified as DMG, H3 K27M-mutant). The authors use gene expression studies, and chromatin profiling to attempt to better understand how these drugs exert an effect on genome regulation. Their main findings are that the drugs reduce cell growth in vitro and in mouse xenografts of patient tumours, that DMG, H3 K27M-mutant tumours are particularly sensitive, identify potential markers of gene expression underlying this sensitivity, and broadly characterize the correlations between chromatin modification changes and gene expression upon treatment, identifying putative pathways that may be affected and underlie the sensitive (and thus how the drugs may affect the tumour cell biology).Strengths:It is a neat, mostly to-the-point work without exploring too many options and possibilities. The authors do a good job not overinterpreting data and speculating too much about the mechanisms, which is a very good thing since the causes and consequences of perturbing such broad epigenetic landscapes of chromatin may be very hard to disentangle. Instead, the authors go straight after testing the performance of the drugs, identifying potential markers and characterizing consequences.Weaknesses:If anything, the experiments done on Figure 3 could benefit from an additional replicate.

We thank the reviewer for evaluating our work, and for the positive and insightful comments.

**Recommendations for the authors:**

**Reviewer #1 (Recommendations For The Authors):**
Perhaps a more substantial drug screen, or CRISPR screen, that utilises single molecule imaging as a readout would identify pharmacologic candidates that are either more effective, or novel.

While out of scope for the current study, this is a highly interesting suggestion, which will be considered in future studies. Here, we focused on the potential use of the novel MYC inhibitor, Sulfopin. While the dependency of DMG cells on MYC signaling has been documented, to the best of our knowledge, pharmacological inhibition of MYC has not been tested for this disease due to the severe lack of potent MYC inhibitors. We show preliminary evidence for the use of this inhibitor, in combination with HDAC inhibition, to attenuate DMG growth *in-vitro* and *in-vivo*.

**Reviewer #2 (Recommendations For The Authors):**
In Figure 1B, it is hard to tell if there are error bars for HSP90 and E2F2. Is there a potential error here? Seems unlikely to not have an error with a RT-qPCR?

We thank the reviewer for the careful evaluation of the figures. We included error bars for all genes shown in Figure 1B. We have now increased the line width with the hope of making this information more accessible. As stated in the figure legend, these error bars represent the standard deviation of two technical repeats.

I noticed that many experiments only had technical replicates. Incorporating biological (independent) replicates, where feasible, would strengthen the study's findings.We agree with the reviewer regarding the importance of biological replicates. While some of the panels present error estimates based on technical repeats, the main results were repeated independently with complementary approaches or various biological systems for validation.

The RNAseq analysis presented in figure 1 was conducted in triplicates and then independently validated by qPCR (Figure 1A-B). Similarly, the transcriptomic analysis presented in figures 2G-I was verified by both western blot (figure 2J) and qPCR (figure S2O). Of note, this later validation was conducted for two different DMG-patient derived cultures.

To verify the robust effects on cellular viability, we analyzed the response to each drug and the combination on eight different DMG-patient-derived cultures, each representing a completely independent experiment. We show very similar trends in response to treatment between cultures that share the same H3-K27M variant. Thus, while for each culture technical repeats are shown, we provide multiple, independent repeats by examining the different cultures. Similarly, in figure 1F we examined the dependency of Sulfopin treatment on the expression of the H3-K27M oncohistone in two independent isogenic systems.

**Reviewer #3 (Recommendations For The Authors):**
A few questions and suggestions:(1) To avoid confusion is important to state if the cells used in each experiment are or not K27M mutants (e.g. SU-DIPG13 on line 63).

We thank the reviewer for pointing this out and have now added this information when appropriate across the manuscript.

1. Line 72 - confirming epigenetic silencing of these genes upon PIN1 inhibition (Fig. 1C, S1D)Considering that the mechanism of down regulation of MYC targets is likely H3K27me3-independent if it is also happening in DMG H3 K27M-mutants (high H3K27me3 here may rather be a consequence of less MYC binding?), I would strike this sentence out and just point out the correlation between lower expression and higher H3K27me3.

We agree with the reviewer that the exact molecular mechanism mediating the silencing is yet to be characterized. We have modified the text in line 72 accordingly.

1. (line 78) Are MYC targets also down regulated in Sulfopin treated DMG, H3 K27M-mutant lines? Any qPCR or previously done RNA-seq data to use?

In addition to the extensive analysis done on SU-DIPG13 cells (Figure 1 and S1), in light of the reviewer`s comment we examined specific MYC targets in an additional H3-K27M mutant DMG culture (SU-DIPG6) treated with Sulfopin, followed by qPCR. We observed a mild reduction in two prominent targets, E2F2 and mTOR (new figure S1D). Unfortunately, within this study, we only conducted full RNA-sequencing analysis on SU-DIPG13 cells treated with Sulfopin, and thus, we could not examine the global effect of Sulfopin on the transcriptome of other DMG cultures. This will, of course, be of high interest for future studies.